# Merging on the Fly Without Retraining: A Sequential Approach to Scalable Continual Model Merging

Anke Tang[1]    Enneng Yang[2]    Li Shen[2*]    Yong Luo[1*]    Han Hu[3]    Lefei Zhang[1]
Bo Du[1]    Dacheng Tao[4]

[1]School of Computer Science, National Engineering Research Center of Multimedia Software
and Hubei Key Laboratory of Multimedia and Network Communication Engineering,
Wuhan University, Wuhan, China
[2]Shenzhen Campus of Sun Yat-sen University, China    [3]Beijing Institute of Technology, Beijing, China
[4]Nanyang Technological University, Singapore
{anketang, luoyong, zhangleifei, dubo}@whu.edu.cn  yangenn@mail.sysu.edu.cn
hhu@bit.edu.cn  {mathshenli, dacheng.tao}@gmail.com

## Abstract

Deep model merging represents an emerging research direction that combines multiple fine-tuned models to harness their specialized capabilities across different tasks and domains. Current model merging techniques focus on merging all available models simultaneously, with weight interpolation-based methods being the predominant approach. However, these conventional approaches are not well-suited for scenarios where models become available sequentially, and they often suffer from high memory requirements and potential interference between tasks. In this study, we propose a training-free projection-based continual merging method that processes models sequentially through orthogonal projections of weight matrices and adaptive scaling mechanisms. Our method operates by projecting new parameter updates onto subspaces orthogonal to existing merged parameter updates while using an adaptive scaling mechanism to maintain stable parameter distances, enabling efficient sequential integration of task-specific knowledge. Our approach maintains constant memory complexity to the number of models, minimizes interference between tasks through orthogonal projections, and retains the performance of previously merged models through adaptive task vector scaling. Extensive experiments on CLIP-ViT models demonstrate that our method achieves a 5-8% average accuracy improvement while maintaining robust performance in different task orderings. Code is publicly available at `https://github.com/tanganke/opcm/`.

## 1  Introduction

With the continued expansion of the scale of foundation models, the demand for efficient model development becomes increasingly pressing [104, 94]. Model merging, as a promising approach to address this challenge, aims to combine multiple fine-tuned models to harness their specialized capabilities in different tasks and domains [43, 92, 50, 35]. Recent studies have demonstrated the effectiveness of model merging in various scenarios, from combining language models with different task expertise [91, 98, 51] to merging vision models trained in distinct domains [86, 73].

The predominant approach to model merging is weight interpolation, which combines model parameters through weighted averaging of the individual models' weights [29, 82]. This method can be formalized as $\theta_{\text{merged}} = \theta^{(0)} + \sum_{i=1}^{n} \alpha_i (\theta^{(i)} - \theta^{(0)})$, where $\theta^{(0)}$ and $\theta^{(i)}$ represent the parameters

---

*Corresponding authors.

39th Conference on Neural Information Processing Systems (NeurIPS 2025).

of the pre-trained model and the $i$-th expert model, respectively, and $\alpha_i$ are the interpolation coefficients, which can be applied task- or layer-wise [52, 96]. Theoretically, the effectiveness of weight interpolation-based merging methods is mainly influenced by two factors: data heterogeneity (the diversity of training set distribution between expert models) and training heterogeneity (differences in hyperparameter settings and optimization dynamics) [59, 78].

However, existing model merging approaches face several fundamental limitations that hinder their practical application and can be revisited from a continual merging perspective. Firstly, current methods typically require concurrent access to all models during the merging process, resulting in significant memory overhead that scales linearly with the number of models being merged [69]. Secondly, the law of commutation in model merging may not hold in continual merging scenarios, where the order of merging models is crucial and can affect the performance of the final merged model. This results in additional challenges for the algorithm design. Third, sequential model availability is a common real-world scenario and lacks attention in existing studies. Last, many current approaches necessitate additional training phases or extensive hyperparameter optimization, introducing substantial computational costs or validation demands [33, 74, 107, 42].

To address these challenges, we propose the Orthogonal Projection-based Continual Merging (OPCM) method which enables a sequential merging as new models become available. Our approach consists of three key features: (1) By maintaining only the current merged and pre-trained models, we sequentially process new models, achieving constant memory complexity $O(|\theta|)$. (2) A projection-based merging mechanism processes parameter updates through orthogonal projections to minimize parameter interference between the incoming and merged models. (3) An adaptive time-varying scaling factor that dynamically adjusts the contribution of each model during the merging process to maintain stable performance, making the method robust to different merging orders. Together, these features enable OPCM to achieve training-free and memory-efficient merging.

Our approach is built upon two key insights: (1) the use of orthogonal projections effectively minimizes interference between different tasks while preserving the specialized capabilities of each model, and (2) an adaptive scaling strategy keeps an approximately constant distance between the merged model and the pre-trained model, ensuring stable performance of the merged model.

In summary, our contributions are (1) We provide a formal mathematical framework for sequential model merging, and discuss its relationship with existing model merging methods. (2) We propose a training-free projection-based continual merging method that processes models sequentially through orthogonal projections and adaptive scaling mechanisms. (3) We conduct extensive experiments on image classification tasks to demonstrate the effectiveness of our method and show 5-8% better accuracy on average while remaining consistently robust regardless of how tasks are ordered.

## 2    Related Work

**Deep Model Fusion** is scalable and effective in various scenarios, including language modeling [106, 79, 81], vision modeling [97, 27], and multimodal modeling [94, 8]. Following [75], deep model fusion techniques can be categorized into three types: model ensemble [68, 3, 48], model merging [43], and model mixing [92, 38]. In this study, we focus on model merging, which does not change the model architecture or introduce inference costs.

(1) *Weight interpolation-based model merging* is one of the most straightforward and intuitive methods for merging model parameters. This approach is largely based on the observation of mode connectivity, which suggests that different solutions within the parameter space can be connected by paths that maintain a consistently low loss [22, 17, 21, 101]. This method assumes that the models being merged lie in similar regions of the loss landscape, allowing for a seamless transition between them without significantly increasing the loss [30, 52, 86, 2, 29, 67]. In addition to its application in multi-task model fusion, weight interpolation can also be utilized during the preference alignment phase in LLM training [103, 46, 7, 25, 61], data-mixing [93, 1], auxiliary-task learning [31], and deep multi-objective optimization [64, 9, 76, 15]. (2) *Alignment-based model merging* represents another approach that focuses on merging models through the reduction of parameter or feature disparities. Current research includes activation and weight matching [72, 33, 95, 36, 90, 54], utilizing channel-wise graph matching [47], and applying permutation invariant [2, 44].

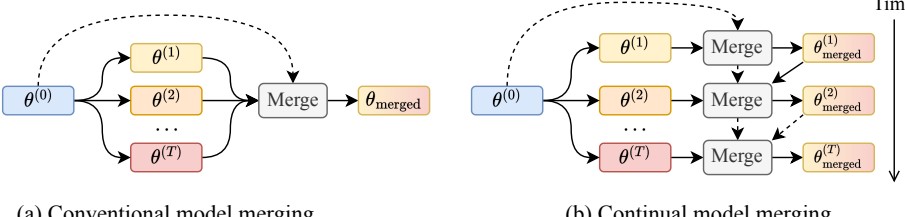

(a) Conventional model merging        (b) Continual model merging

Figure 1: Comparison between conventional and continual model merging approaches. (a) Conventional model merging requires simultaneous access to all expert models, performing merging in a single step. (b) Continual model merging processes models sequentially as they become available.

However, the above methods require all models to be available simultaneously for merging, which limits their applicability. In contrast, our approach enables the continual merging of sequentially arriving models with higher memory efficiency.

**Continual Learning** addresses scenarios where models must sequentially learn new tasks while maintaining performance on previously acquired knowledge [84, 105]. This challenging paradigm has led to several key approaches: memory-based methods that store past samples [49, 6, 5], architecture-based methods that modify network structures [66, 20, 99], regularization-based methods involve the addition of regularization loss terms [37, 102], subspace-based methods that use lower-dimensional projections [19, 16], and Bayesian methods via incorporating uncertainty estimation [56]. However, the aforementioned methods are all based on learning from the original data. In contrast, the continual merging approach proposed in this paper allows us to directly merge a sequence of models (i.e., tasks) arriving in a stream, offering an orthogonal perspective to continual learning.

## 3 Rethinking Model Merging From a Continual Learning Perspective

In this section, we present a novel perspective on multi-task model merging through the lens of continual learning, wherein task-specific models are obtained sequentially rather than simultaneously.

### 3.1 Problem Setup

Formally, let $f_{\theta^{(0)}} : \mathcal{X} \rightarrow \mathcal{Y}$ denote the pre-trained model parameterized by $\theta^{(0)}$, where $\mathcal{X}$ represents the input space and $\mathcal{Y}$ represents the output space. Consider $T$ task-specific models: $\{f_{\theta^{(1)}}, f_{\theta^{(2)}}, \ldots, f_{\theta^{(T)}}\}$, each fine-tuned from the pre-trained model $f_{\theta^{(0)}}$ using the corresponding data set $D_i$ for task $i$. Each model may incorporate a task-specific head $h_{\phi_i} : \mathcal{Y} \rightarrow \mathcal{Z}_i$. Our objective is to construct a unified model $f_{\theta_{\text{merged}}} : \mathcal{X} \rightarrow \mathcal{Y}$ that maintains high performance in all $T$ tasks while preserving task-specific heads $\{h_{\phi_i}\}_{i=1}^{T}$. Initially, we define the conventional versus continual model merging approaches in formal terms and subsequently explore the challenges and current solutions from a continual learning perspective. In Figure 1, we provide a visual comparison between conventional and continual model merging techniques, highlighting their distinctive approaches.

**Definition 3.1** (Conventional Model Merging). Conventional model merging approaches predominantly address scenarios where all expert models are available simultaneously, executing the merge operation in a single step. This process can be formally expressed as follows:

$$\theta_{\text{merged}} = \text{Merge}(\theta^{(0)}; \theta^{(1)}, \theta^{(2)}, \ldots, \theta^{(T)}). \tag{1}$$

**Definition 3.2** (Continual Model Merging). Task-specific models are introduced progressively over time in the continuous model merging scenario. Model merging occurs incrementally in this paradigm as each new task-specific model becomes available. Specifically, at step $t$, we merge the current merged model parameterized by $\theta_{\text{merged}}^{(t-1)}$ with the newly trained task-specific model parameterized by $\theta^{(t)}$, yielding an updated merged model parameterized by $\theta_{\text{merged}}^{(t)}$. This process can be formalized as:

$$\theta_{\text{merged}}^{(t)} = \text{ContinualMerge}(\theta_{\text{merged}}^{(t-1)}; \theta^{(0)}, \theta^{(t)}), \ t \geq 2, \tag{2}$$

where typically $\theta_{\text{merged}}^{(1)} = \theta^{(1)}$ is initialized as the first expert model and $\theta_{\text{merged}}^{(t)}$ aims to minimize the expected risk $\theta_{\text{merged}}^{(t)} = \arg\min_\theta \mathbb{E}_{(x,y) \sim \mathcal{D}_1 \cup \ldots \mathcal{D}_t} \mathcal{L}_t\left(f_\theta(x), y\right)$.

## 3.2 Opportunities & Challenges in Continual Merging

**Storage and memory efficiency** is a significant advantage of continual merging over traditional merging methods. In continual merging, at each step $t$, only a fixed number of models need to be stored: the current merged model, the new model to be merged, and optionally the pre-trained base model. This approach leads to a constant memory complexity of $O(|\theta|)$, where $|\theta|$ represents the size of an individual model. Crucially, this memory requirement remains constant, regardless of the total number of downstream tasks $T$ involved. In contrast, as shown in Figure 2, conventional merging methods typically involve maintaining all models, which results in a linear memory complexity of $O(T|\theta|)$.

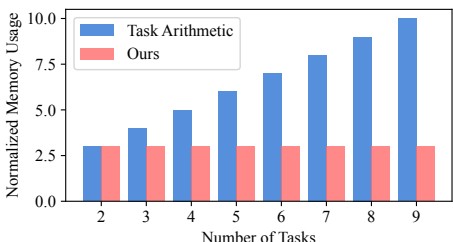

Figure 2: Memory complexity of task arithmetic [29] and our method.

**The law of commutation in model merging** refers to the property that the sequence in which models are combined does not influence the outcome of the merged model. Formally, this commutative property is described by the equation for standard merging techniques as the following holds:

$$\text{Merge}(\theta^{(0)}; \theta^{(i)}, \theta^{(j)}) = \text{Merge}(\theta^{(0)}; \theta^{(j)}, \theta^{(i)}), \ \forall i, j. \tag{3}$$

However, in continual model merging, this commutative characteristic *may not hold*. Here, the ordering of models can have an impact on the final performance of the merged model. This is because, in a continual merging setting, each step of merging builds upon the previously merged model, resulting in sequential dependencies that disrupt commutativity. To illustrate, consider the continual merging process ContinualMerging (CM), where the commutative property may not hold:

$$\text{CM}\left(\text{CM}\left(\theta_{\text{merged}}^{(i-1)}; \theta^{(0)}, \theta^{(i)}\right); \theta^{(0)}, \theta^{(i+1)}\right) \stackrel{?}{=} \text{CM}\left(\text{CM}\left(\theta_{\text{merged}}^{(i-1)}; \theta^{(0)}, \theta^{(i+1)}\right); \theta^{(0)}, \theta^{(i)}\right), \ \forall i. \tag{4}$$

This non-commutative nature introduces additional complexities in the knowledge preservation of preceding models, where the quality of the final merged model hinges not just on the models themselves but critically on the ordering in which they are integrated. Moreover, *as earlier parameter information is gradually diminished throughout the sequential process, the final model might not completely encapsulate the knowledge from all individual tasks*. Consequently, it becomes essential to develop effective merging strategies that take the non-commutation property into account, balancing sequential dependencies and task knowledge retention.

**Existing applications of continual model merging** are prevalent in online training settings, where models are updated along the training trajectory. This approach includes techniques such as the latest weight averaging (LAWA) [34], exponential moving average (EMA) [53, 65], and stochastic weight averaging (SWA) [30]. These strategies, however, were not originally intended for scenarios with significant distribution shifts between consecutive models, which can be a critical challenge. In Dziadzio et al. [18], the authors propose the concept of temporal model merging, where they select models to merge at each step, akin to continual model merging. While most existing model merging approaches assume the simultaneous availability of expert models, research on continual model merging remains limited [43]. Some conventional model merging techniques can be revised and adapted within the framework of continual model merging. As an example, consider task arithmetic:

*Baseline* 3.3 (Continual Task Arithmetic). This is a straightforward extension of the conventional task arithmetic to the continual model merging scenario. The update rule can be expressed as follows:

$$\theta_{\text{merged}}^{(t)} = \theta_{\text{merged}}^{(t-1)} + \lambda(\theta^{(t)} - \theta^{(0)}), \tag{5}$$

where $\lambda$ is a scalar hyperparameter that controls the contribution of the new task-specific model.

To summarize, continual model merging methods align with real-world applications where tasks emerge progressively and offer improved memory efficiency. However, it can be non-commutative, which introduces unique challenges in task knowledge retention and interference resolution.

## 4 Methodology

Our approach to continual model merging is motivated by three key insights: (1) *Orthogonality Preservation:* To minimize task interference, the parameter updates for new tasks should be approximately orthogonal to the existing merged parameter updates. (2) *Magnitude Control:* The magnitude

of parameter changes should remain stable throughout the merging process to prevent drift from the pre-trained model's loss basin. (3) *Information Retention:* The merged model should preserve task-specific information from all previously seen models while accommodating new tasks efficiently.

Building on these insights, we propose a projection-based continual merging approach that (1) Projects new task vectors onto subspaces orthogonal to the current merged model; (2) Employs adaptive time-varying scaling to maintain stable parameter drift from the base model; (3) Preserves task-specific information through careful parameter updates.

The complete procedure is outlined in Algorithm 1 and illustrated in Figure 3. Starting with the first task-specific model $f_{\theta^{(1)}}$, our method iteratively incorporates new models as they become available. For each new model, we handle the parameters in two distinct ways: weight matrices in linear layers undergo orthogonal projection to minimize interference, whereas other parameters, such as biases and those in non-linear layers, are averaged directly.

**An overview of the update rule** is provided below. For weight matrices in linear layers, we propose the following update rule:

$$W_{\text{merged}}^{(t)} = W^{(0)} + \frac{\lambda^{(t-1)}\Delta W_{\text{merged}}^{(t-1)} + \mathcal{P}_{\alpha}^{(t-1)}\left(\Delta W^{(t)}\right)}{\lambda^{(t)}},$$
(6)

where $W \in \mathbb{R}^{m \times n}$, $\Delta W_{\text{merged}}^{(t-1)} = W_{\text{merged}}^{(t-1)} - W^{(0)}$ and $\Delta W^{(t)} = W^{(t)} - W^{(0)}$ denote the element-wise differences between the model parameters and the pre-trained model, commonly referred to as task vectors or delta parameters [29, 100]. $\mathcal{P}_{\alpha}^{(t-1)}(\cdot)$ represents a projection mapping that projects the task vector $\Delta W^{(t)}$ onto the subspace spanned by $\Delta W_{\text{merged}}^{(t-1)}$. When a new task model (shown in solid red arrow in Figure 3) arrives, its difference from the pre-trained model $\theta^{(0)}$ is projected onto this orthogonal subspace using the projection operator $\mathcal{P}_{\alpha}^{(t-1)}$. $\lambda^{(t-1)}$ denotes a time-varying scaling factor that controls the contribution of the projected task vector to the merged model. By induction, we can expand Eq.(6) to obtain the general term formula, proved in Theorem A.1:

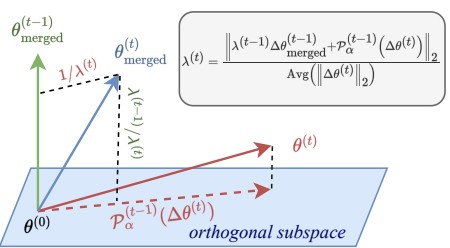

Figure 3: **Subspace view.** Geometric interpretation of the proposed continual merging approach, illustrating the orthogonal projection and adaptive scaling mechanisms.

---

**Algorithm 1** OPCM

---

1: Initialize $\theta_{\text{merged}}^{(1)} = \theta^{(1)}$, average norm of the task vectors $n = \|\Delta\theta^{(1)}\|_2$, scaling factor $\lambda^{(1)} = 1$.
2: **for** $t = 2$ to $T$ **do**
3:     **for** weight matrices $W \in \mathbb{R}^{m \times n}$ in linear layers **do**
4:         $\Delta W_{\text{merged}}^{(t-1)} \leftarrow W_{\text{merged}}^{(t-1)} - W^{(0)}$
5:         $\Delta W^{(t)} \leftarrow W^{(t)} - W^{(0)}$
6:         $\Delta W_{\text{proj}}^{(t)} \leftarrow \mathcal{P}_{\alpha}^{(t-1)}\left(\Delta W^{(t)}; \Delta W_{\text{merged}}^{(t-1)}\right)$
7:     **end for**
8:     **for** other parameters $p$ **do**
9:         $\Delta p^{(t)} \leftarrow p^{(t)} - p^{(0)}$
10:    **end for**
11:    $\Delta\theta_{\text{merged}}^{(t-1)} \leftarrow \theta_{\text{merged}}^{(t-1)} - \theta^{(0)}$
12:    Concatenate $\Delta W_{\text{proj}}^{(t)}$ and $\Delta p^{(t)}$ into $\Delta\theta_{\text{proj}}^{(t)}$
13:    $n \leftarrow \frac{t-1}{t}n + \frac{1}{t}\|\Delta\theta^{(t)}\|_2$
14:    $\lambda^{(t)} \leftarrow \left\|\lambda^{(t-1)}\Delta\theta_{\text{merged}}^{(t-1)} + \Delta\theta_{\text{proj}}^{(t)}\right\|_2 / n$
15:    $\theta_{\text{merged}}^{(t)} \leftarrow \theta^{(0)} + \frac{\lambda^{(t-1)}\Delta\theta_{\text{merged}}^{(t-1)} + \Delta\theta_{\text{proj}}^{(t)}}{\lambda^{(t)}}$
16: **end for**
17: **return** $\theta_{\text{merged}}^{(T)}$

---

$$W_{\text{merged}}^{(t)} = W^{(0)} + \frac{1}{\lambda^{(t)}}\sum_{i=1}^{t}\mathcal{P}_{\alpha}^{(i-1)}\left(\Delta W^{(i)}\right).$$
(7)

This formulation can be interpreted as a more generalized form of Task Arithmetic, where the task vectors are projected onto a subspace spanned by the previous merged model.

For biases in linear layers and other parameters, we simply set $\mathcal{P}_{\alpha}^{(t-1)}$ in Eq.(6) as the identity mapping. Thus the update rule is simplified as $p_{\text{merged}}^{(t)} = p^{(0)} + (\lambda^{(t-1)}\Delta p_{\text{merged}}^{(t-1)} + \Delta p^{(t)})/\lambda^{(t)}$. The general term formula degenerates to a similar form as Task Arithmetic, i.e., $p_{\text{merged}}^{(t)} = p^{(0)} + \frac{1}{\lambda^{(t)}}\sum_{i=1}^{t}\Delta p^{(i)}$.

**The projection mapping** $\mathcal{P}_{\alpha}^{(t-1)}$ is designed to ensure orthogonality between the incoming task vector $\Delta W^{(t)}$ and the previous merged model $\Delta W_{\text{merged}}^{(t-1)}$ while maintaining proximity between $\mathcal{P}_{\alpha}^{(t-1)}\left(\Delta W^{(t)}\right)$ and $\Delta W^{(t)}$. We first perform a full singular value decomposition (SVD) on the previous merged model $\Delta W_{\text{merged}}^{(t-1)} = U^{(t-1)}\Sigma^{(t-1)}V^{(t-1)}$, where $U^{(t-1)} \in \mathbb{R}^{m \times m}$ and $V^{(t-1)} \in$

$\mathbb{R}^{n \times n}$ comprise orthonormal basis vectors, and $\Sigma^{(t-1)} \in \mathbb{R}^{m \times n}$ is a (rectangular) diagonal matrix containing the singular values [58]. The proposed projection mapping $\mathcal{P}_\alpha^{(t-1)}$ is then defined as: $\mathcal{P}_\alpha^{(t-1)}\left(\Delta W^{(t)}\right) = \sum_{i,j=r_\alpha, i \neq j}^{m,n} \left\langle \Delta W^{(t)}, u_i v_j^T \right\rangle_F u_i v_j^T$, where $u_i$ and $v_j$ denote the $i$-th column of $U^{(t-1)}$ and the $j$-th column of $V^{(t-1)}$, respectively. $\alpha$ is the projection threshold hyper-parameter, which controls the balance between retaining existing knowledge and incorporating knowledge from new tasks. $r_\alpha$ represents the minimal rank of $\Delta W_{\text{merged}}^{(t-1)}$ such that $\sum_{i=1}^{r_\alpha} \sigma_i \geq \alpha \sum_{i=1}^{\min(m,n)} \sigma_i$.

**The adaptive time-varying scaling factor** $\lambda^{(t)}$ is introduced to maintain a consistent magnitude of the merged model's deviation from the pre-trained model throughout the merging process, specifically ensuring that the Frobenius norm $\|W_{\text{merged}}^{(t)} - W^{(0)}\|_2$ remains stable over time. The scaling factor $\lambda^{(t)}$ is computed as $\lambda^{(1)} = 1$, $\lambda^{(t)} = \left\|\lambda^{(t-1)}\Delta\theta_{\text{merged}}^{(t-1)} + \mathcal{P}_\alpha^{(t-1)}\left(\Delta\theta^{(t)}\right)\right\|_2 / \text{Avg}\left(\left\|\Delta\theta^{(t)}\right\|_2\right)$, where $\Delta\theta_{\text{merged}}^{(t-1)} = \theta_{\text{merged}}^{(t-1)} - \theta^{(0)}$ and $\Delta\theta^{(t)} = \theta^{(t)} - \theta^{(0)}$ represent flattened vectors of parameter differences, and $\text{Avg}(\|\Delta\theta^{(t)}\|_2) = \frac{1}{t}\sum_{i=1}^{t}\|\Delta\theta^{(i)}\|_2$ is the average norm. For notational simplicity, we apply $\mathcal{P}_\alpha^{(t-1)}$ directly to the task vector rather than to individual weight matrices.

Alternatively, the time-varying scaling factor can be set to $\lambda^{(t)} = \sqrt{t}$. This choice is motivated by empirical observations from prior work indicating that task vectors from different tasks tend to be approximately orthogonal [29, 77, 32], i.e., $\left\langle \Delta\theta^{(i)}, \Delta\theta^{(j)} \right\rangle \approx \delta_{ij}$ generally holds for all $i, j \in [1, t]$. The empirical validation of task vector orthogonality is presented in Figure 4, which analyzes 8 task-specific ViT-B/32 models. For a more comprehensive analysis across model sizes and task quantities, we present additional results in the appendix: Figure 9 (ViT-B/32), Figure 10 (ViT-B/16), and Figure 11 (ViT-L/14), each evaluated on 20 downstream tasks. This orthogonality property makes $\sqrt{t}$ a natural choice for the scaling factor, as it helps maintain the magnitude of parameter changes across merging steps.

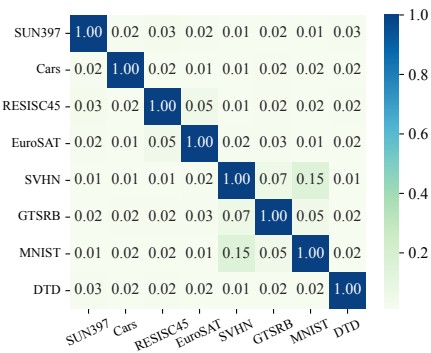

Figure 4: The cosine similarity between task vectors of ViT-B/32.

## 5 Theoretical Analysis

We now present some theoretical analysis of the proposed continual merging method, establishing key properties regarding orthogonality and stability. Proofs are provided in Appendix A.1 and additional discussion on the computational complexity of OPCM can be found in Appendix B.

**Theorem 5.1** (Orthogonality of Projected Task Vectors). *Given a sequence of task-specific models* $\{f_{\theta^{(t)}}\}_{t=1}^{T}$ *fine-tuned from a pre-trained model* $f_{\theta^{(0)}}$, *for any time step* $t$, *the projected task vector* $\mathcal{P}_\alpha^{(t-1)}(\Delta W^{(t)})$ *obtained from the update rule in Eq.(6) is orthogonal to* $\Delta W_{\text{merged}}^{(t-1)}$, *i.e.,*

$$\left\langle \mathcal{P}_\alpha^{(t-1)}\left(\Delta W^{(t)}\right), \Delta W_{\text{merged}}^{(t-1)} \right\rangle_F = 0. \tag{8}$$

Similar to [89], we can define the merging gap for $i$-th task as follows:

$$G_i = \mathcal{L}_i\left(\theta^{(0)} + \sum_{j=1}^{t} \eta_j \mathcal{P}_\alpha^{(j-1)}\left(\Delta\theta^{(i)}\right)\right) - \mathcal{L}_i\left(\theta^{(0)} + \eta_i \mathcal{P}_\alpha^{(i-1)}\left(\Delta\theta^{(i)}\right)\right). \tag{9}$$

Where $\{\eta_i\}_{i=1}^{t}$ is a sequence of scaling factors, and $\mathcal{L}_i$ is the loss function for $i$-th task. Apply Taylor expansion to $G_i$ at $\theta^{(0)}$, we obtain the first-order approximation $G_i \approx \nabla_{\theta^{(0)}}\mathcal{L}_i^\top\left(\sum_{j=1, j \neq i}^{t} \eta_j \mathcal{P}_\alpha^{(j-1)}\left(\Delta\theta^{(i)}\right)\right)$. At time step $t$, we have $\eta_i = 1/\lambda^{(t)}$ for all $i \in [1, t]$. $G_t$ is the merging gap for $t$-th task, which can be simplified as $G_t \approx \nabla_{\theta^{(0)}}\mathcal{L}_t^\top\left(\frac{\lambda^{(t-1)}}{\lambda^{(t)}}\Delta\theta_{\text{merged}}^{(t-1)}\right)$.

On the other hand, since the task vector $\Delta\theta^{(i)}$ represents the cumulative effect of gradient updates throughout the training process, we can express the merging gap in terms of the inner product between

Table 1: The performance comparison of continual merging methods. We report the average accuracy and backward transfer (BWT) of the merged models. The best results are highlighted in bold. We abbreviate 'Continual' as 'C.' in the table to save space.

| | | ViT-B/32 | | | ViT-B/16 | | | ViT-L/14 | | |
| | Method | 8 tasks | 14 tasks | 20 tasks | 8 tasks | 14 tasks | 20 tasks | 8 tasks | 14 tasks | 20 tasks |
|---|---|---|---|---|---|---|---|---|---|---|
| | Pre-Trained | 48.1 | 56.9 | 55.6 | 55.4 | 62.0 | 59.8 | 64.9 | 69.1 | 65.6 |
| | Fine-Tuned | 90.4 | 89.3 | 89.8 | 92.4 | 91.3 | 91.6 | 94.3 | 93.4 | 93.5 |
| | C. Fine-Tuned | 79.8 | 67.4 | 62.6 | 82.9 | 72.2 | 68.2 | 90.0 | 70.9 | 77.7 |
| ACC↑ | Average (SWA) | 66.3±0.0 | 65.4±0.0 | 61.1±0.0 | 72.3±0.0 | 69.7±0.0 | 64.8±0.0 | 80.0±0.0 | 77.5±0.0 | 71.1±0.0 |
| | C. Task Arithmetic | 67.5±0.0 | 66.5±0.0 | 60.6±0.0 | 77.1±0.0 | 70.9±0.0 | 64.2±0.0 | 82.1±0.0 | 77.9±0.0 | 70.3±0.0 |
| | C. Ties-Merging | 49.0±10.2 | 66.2±0.6 | 59.9±0.7 | 66.8±3.7 | 70.5±0.8 | 63.0±1.6 | 64.3±7.0 | 78.0±0.6 | 68.3±0.9 |
| | **OPCM (Ours)** | **75.5**±0.5 | **71.9**±0.3 | **65.7**±0.2 | **81.8**±0.3 | **77.1**±0.5 | **70.3**±0.2 | **87.0**±0.4 | **83.5**±0.2 | **76.0**±0.2 |
| BWT↑ | Average (SWA) | -11.5±2.2 | -8.0±1.3 | -7.1±2.1 | -9.7±1.5 | -7.1±1.4 | -7.3±1.7 | -7.3±1.4 | -5.8±1.0 | -6.4±1.5 |
| | C. Task Arithmetic | -9.6±1.5 | -1.3±0.3 | -3.4±0.4 | **-4.2**±1.0 | -1.3±0.4 | -3.6±0.4 | -7.1±0.8 | -1.8±0.3 | -3.3±0.3 |
| | C. Ties-Merging | -15.3±8.0 | **1.9**±0.6 | **-1.5**±0.7 | -5.5±0.4 | **1.4**±0.7 | **-1.5**±1.2 | -13.0±5.7 | **1.1**±0.4 | **-2.9**±1.0 |
| | **OPCM (Ours)** | **-6.3**±1.1 | -6.0±1.0 | -7.8±1.5 | -4.8±0.7 | -5.1±1.4 | -6.3±2.2 | **-2.6**±1.0 | -4.3±0.7 | -6.5±1.8 |

task vectors as $G_t \approx C\Delta\theta^{(t)^\top}\Delta\theta_{\text{merged}}^{(t-1)}$, where $C$ is a constant [77, 89]. Consequently, enforcing orthogonality between these vectors ($\Delta\theta^{(t)^\top}\Delta\theta_{\text{merged}}^{(t-1)} = 0$) serves to minimize the gap $G_t$.

**Theorem 5.2** (Bounded Parameter Distance). *Under the update rule in Eq.(6) with the empirical choice of scaling factor $\lambda^{(t)} = \sqrt{t}$, the distance between the merged model and the pre-trained model remains bounded:*

$$\left\|W_{merged}^{(t)} - W^{(0)}\right\|_F^2 \le \max_{i\in[1,t]}\left\|\Delta W^{(i)}\right\|_F^2 . \tag{10}$$

The proposed continual merging method ensures that the merged model maintains a stable distance from the pre-trained model throughout the merging process, preventing parameter drift that could lead to catastrophic forgetting.

**Corollary 5.3** (Preservation of Task Information). *For any time step t, the merged model preserves information from all previous tasks in the following sense:*

$$W_{merged}^{(t)} - W^{(0)} = \frac{1}{\lambda^{(t)}}\sum_{i=1}^{t}\mathcal{P}_\alpha^{(i-1)}\left(\Delta W^{(i)}\right) . \tag{11}$$

This formulation follows directly from Eq.(7). This corollary demonstrates that our continual merging approach maintains a weighted combination of all previously learned task-specific changes, where each task's contribution is preserved through its projection. In Appendix A.2, we further discuss the special case where two adjacent task vectors are highly correlated, and our method can automatically filter redundant knowledge in the incoming task vector.

# 6 Experiments

In this section, we present the experimental results of our continual merging approach and the ablation studies for a comprehensive analysis of our method.

**Experimental Setup**: (1) *Models and datasets*. We first fine-tune the CLIP-ViT models [63] on up to 20 downstream tasks. Following [83], we evaluate our approach on three task sets of increasing size: a base set of 8 tasks, an extended set of 14 tasks, and the complete set of 20 tasks. This hierarchical grouping allows us to analyze how our method scales with increasing task diversity and complexity. Additional experimental details are provided in Appendix C. (2) *Evaluation metrics*. To evaluate our merged models, we employ two key metrics: average accuracy (ACC) and backward transfer (BWT) [45], details are provided in Appendix D.2.

## 6.1 Continual Multi-Task Model Merging

We evaluate our method on three different CLIP-ViT architectures (ViT-B/32, ViT-B/16, and ViT-L/14) across three task sets of increasing size (8, 14, and 20 tasks). For each experiment, we set $\alpha = 0.5$ and repeat the experiment 10 times with shuffled task order and report the mean and standard deviation of the results. We compare our continual merging approach against four baselines: (1) Continual fine-tuning, (2) Simple Weight Averaging (SWA), which takes the arithmetic mean of model parameters, (3) Continual Task Arithmetic [29], which performs weighted averaging based on task-specific scaling factors, and (4) Continual Ties-Merging [91]. Details are in Appendix D.3.1.

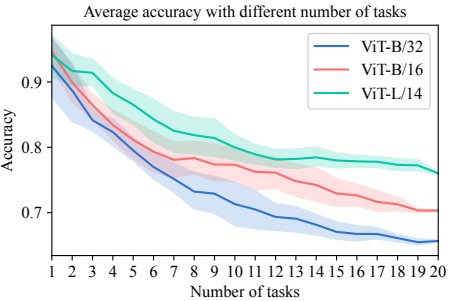

Figure 5: Performance comparison of ViT models with different architectures across an increasing number of sequential tasks.

As shown in Table 1, our method consistently outperforms the baseline methods across all model architectures and task sets. Our method consistently achieves improvements of 5-8% over continual Task Arithmetic and continual Ties-Merging. In addition, the performance gain is maintained as we scale the task set size and model capacity. Furthermore, the backward transfer (BWT) results demonstrate our method's ability to retain knowledge of previously learned tasks. While some negative transfer is inevitable in a continual learning setting, our approach shows better BWT scores compared to the baselines in most cases. The larger ViT-L/14 model exhibits the least forgetting, with a BWT of only -2.6% on the 8-task set, suggesting that increased model capacity helps mitigate catastrophic forgetting.

In Figure 5, we show a line plot comparing the average accuracy of three different Vision Transformer (ViT) architectures as they handle an increasing number of tasks from 1 to 20 ($\alpha = 0.5$). For each run, the order of the tasks is shuffled. Each line represents a different model variant, with shaded areas indicating confidence intervals. The plot demonstrates a general downward trend in accuracy as more tasks are added, with ViT-L/14 maintaining the highest performance throughout. Starting from around 90% accuracy for all models, there's a gradual decline, with ViT-L/14 maintaining approximately 75-80% accuracy by task 20, while the smaller models (ViT-B/32 and ViT-B/16) show steeper degradation, dropping to around 65-70% accuracy.

(1) *Robustness to task ordering.* The relatively narrow confidence bands in Figure 5 and small standard deviations in Table 1 demonstrate that our method is robust to different task orderings, maintaining consistent overall performance regardless of the sequence in which tasks are presented. Even as the number of tasks increases to 20, the performance variation remains contained, suggesting that our orthogonal projection mechanism effectively manages task interference regardless of the order in which tasks are merged. This robustness to task order is a crucial practical advantage of a continual merging technique, as it eliminates the need for careful task sequence engineering in real-world applications. (2) *Model size matters in average performance.* We also observe that model size plays a crucial role in the merging performance. The ViT-L/14 consistently achieves better results than ViT-B/32 and ViT-B/16, this indicates that larger models may be better suited for continual multi-task merging, possibly due to their increased dimensionality, which helps manage parameter interference by expanding the orthogonal subspace where task-specific updates can exist without affecting other tasks. This property is particularly beneficial when merging multiple task-specific models. (3) *Model size matters in mitigating forgetting.* The results in Table 1 and Figure 5 also indicate that model size plays a crucial role in mitigating catastrophic forgetting during continual merging. While all models show some degree of performance degradation as the number of tasks increases, the rate of decline varies significantly with model capacity. The ViT-L/14, our largest model, exhibits the most graceful degradation (-17.5% drops at 20 tasks compared to individuals). In contrast, the smaller ViT-B/32 and ViT-B/16 models show steeper performance drops (24.1% and 21.3% average accuracy drops at 20 tasks, respectively).

## 6.2 Hyper-Parameter Analysis

*Ablations on the projection threshold $\alpha$.* We perform ablation studies on the projection threshold $\alpha$ to validate the robustness of proposed method to different hyper-parameter settings. Figure 6 presents a comparative ablation study between Task Arithmetic (a) and our method (b) on ViT-B/32 across

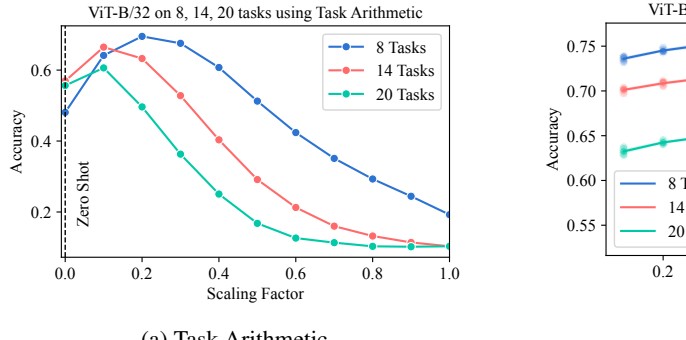

(a) Task Arithmetic

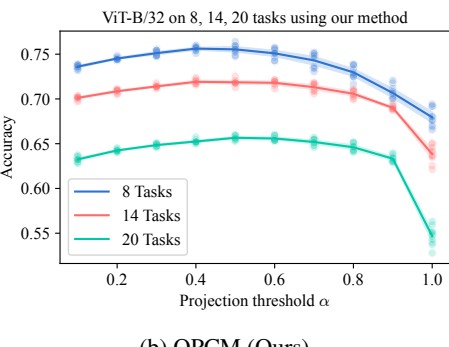

(b) OPCM (Ours)

Figure 6: Ablations on the hyper-parameter settings. We compare our method with Task Arithmetic. This figure highlights the robustness of our method to different hyper-parameter settings.

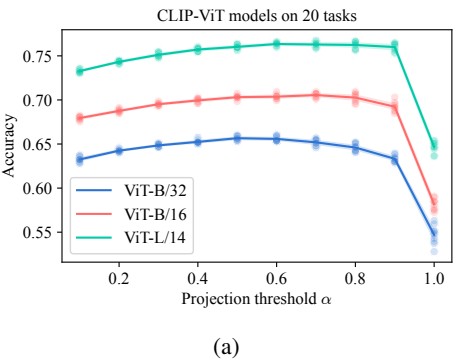

(a)

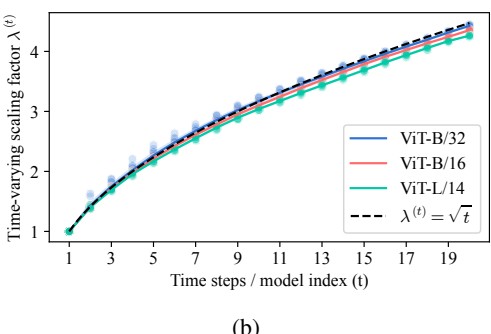

(b)

Figure 7: Hyper-parameters analysis. (a) Performance comparison with varying size of models across different projection threshold on 20 tasks; (b) The adaptive scaling factor $\lambda^{(t)}$ across different steps.

different task scales. Several key findings emerge from this comparison: (1) *Knowledge retention and new task learning trade-off*. Across all task sets, the performance curves exhibit a consistent inverted U-shape pattern, with optimal performance achieved in the range of $\alpha \approx 0.4 - 0.6$. This suggests that moderate projection thresholds strike an optimal balance between preserving task-specific information and managing interference between tasks. (2) *Hyperparameter robustness and transferability*. Our method demonstrates remarkably more stable performance across hyperparameter ranges compared to Task Arithmetic. The optimal $\alpha$ value remains relatively consistent ($\approx 0.5$) regardless of the number of tasks, demonstrating the robustness and transferability across different configurations. This stability and transferability is particularly valuable from a practical perspective especially when hyper-parameter tuning is expensive, thus minimal tuning of $\alpha$ is required when scaling to different numbers of tasks. In Figure 7a we also show the performance comparison of CLIP-ViT models of different sizes with varying $\alpha$ on 20 tasks. It is observed that all three models show stable performance and similar trends as the projection threshold changes. More details are provided in Appendix D.4.

*Empirical analysis of the adaptive time-varying scaling factor $\lambda^{(t)}$.* In our method, the scaling factor $\lambda^{(t)}$ is adaptively adjusted based on the norm ratio between the combined task vectors and the average task vector norm. This adaptive mechanism helps maintain stable parameter distances throughout the merging process. Figure 7b shows the adaptive scaling factor $\lambda^{(t)}$ across different steps. The adaptive scaling factors closely follow a $\sqrt{t}$ curve (shown as the dashed black line), empirically validating our hypothesis of an empirical choice of $\lambda^{(t)} = \sqrt{t}$ in Section 4. All three model architectures (ViT-B/32, ViT-B/16, and ViT-L/14) exhibit remarkably similar scaling patterns, indicating that this time-varying adaptive scaling mechanism is also robust and transferable across different model capacities.

# 7 Conclusion and Future Work

In this work, we presented a novel training-free projection-based continual model merging method for combining multiple fine-tuned models *sequentially*. Our approach addresses several key challenges in model merging through orthogonal projections of weight matrices and adaptive scaling mechanisms.

Through extensive experiments, we demonstrated that our method consistently outperforms baseline approaches. The results show that our approach achieves 5-8% average accuracy improvement while maintaining robust performance across different task orderings. The effectiveness of our method scales well with model capacity, with larger models showing a better ability to mitigate catastrophic forgetting during sequential merging.

While our current work focuses on vision models, the proposed method's principles are generally applicable to other domains. Future work could explore extensions to language models, multi-modal architectures, and other scenarios where sequential model merging is beneficial. Additionally, investigating the relationship between model capacity and merging performance could provide valuable insights.

## Broader impacts

This research advances the field of model merging with potential positive impacts on computational efficiency and accessibility of machine learning systems. By enabling effective continual merging of task-specific models, our method can significantly reduce storage requirements and computational costs associated with maintaining multiple specialized models. This efficiency gain could lead to reduced energy consumption and carbon footprint in AI deployments.

## Acknowledgments and Disclosure of Funding

This work is supported by the National Natural Science Foundation of China (Grant No. 62225113, U23A20318, U2336211, 62576364 and 62276195), the Foundation for Innovative Research Groups of Hubei Province (Grant No. 2024AFA017), the Science and Technology Major Project of Hubei Province (Grant No. 2024BAB046) and the Shenzhen Basic Research Project (Natural Science Foundation) Basic Research Key Project (NO. JCYJ20241202124430041). Dr. Tao's research is partially supported by NTU RSR and Start Up Grants. The numerical calculations in this paper have been done on the supercomputing system in the Supercomputing Center of Wuhan University.

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

# A  Additional Theoretical Analysis

## A.1  Proofs of Theorems

In this section, we provide detailed proofs for the theorems and corollaries presented in the main text.

First, we prove the general term formula of the merged model, which demonstrates that the merged model can be expressed as a weighted combination of all previous task-specific models. This formula shows how each task's contribution is preserved through its projection operator and serves as the foundation for understanding the key properties of our continual merging approach.

**Theorem A.1** (General Term Formula). *Given the update rule of our method as defined in Eq.(6), the general term formula of the merged model at time step $t$ can be expressed as:*

$$W_{merged}^{(t)} = W^{(0)} + \frac{1}{\lambda^{(t)}} \sum_{i=1}^{t} \mathcal{P}_{\alpha}^{(i-1)}\left(\Delta W^{(i)}\right). \tag{12}$$

*Proof.* We prove the general term formula by mathematical induction. For the base case $t = 1$, we have $W_{merged}^{(1)} = W^{(1)} = W^{(0)} + \frac{\mathcal{P}_{\alpha}^{(0)}(\Delta W^{(1)})}{\lambda^{(1)}}$, where $\mathcal{P}_{\alpha}^{(0)}$ is the identity mapping and $\lambda^{(1)} = 1$. For $t = 2$, applying the update rule yields:

$$W_{merged}^{(2)} = W^{(0)} + \frac{\lambda^{(1)}\Delta W_{merged}^{(1)} + \mathcal{P}_{\alpha}^{(1)}\left(\Delta W^{(2)}\right)}{\lambda^{(2)}}, \tag{13}$$

$$= W^{(0)} + \frac{\lambda^{(1)}\frac{\mathcal{P}_{\alpha}^{(0)}\left(\Delta W^{(1)}\right)}{\lambda^{(1)}} + \mathcal{P}_{\alpha}^{(1)}\left(\Delta W^{(2)}\right)}{\lambda^{(2)}}, \tag{14}$$

$$= W^{(0)} + \underbrace{\frac{\mathcal{P}_{\alpha}^{(0)}\left(\Delta W^{(1)}\right) + \mathcal{P}_{\alpha}^{(1)}\left(\Delta W^{(2)}\right)}{\lambda^{(2)}}}_{\Delta W_{merged}^{(2)}}. \tag{15}$$

Thus, the formula holds for $t \leq 2$. For the inductive step, assume the formula holds for $t = k - 1$. Then for $t = k$:

$$W_{\text{merged}}^{(k)} = W^{(0)} + \frac{\lambda^{(k-1)} \Delta W_{\text{merged}}^{(k-1)} + \mathcal{P}_\alpha^{(k-1)} \left( \Delta W^{(k)} \right)}{\lambda^{(k)}}, \tag{16}$$

$$= W^{(0)} + \frac{\lambda^{(k-1)} \left( \frac{1}{\lambda^{(k-1)}} \sum_{i=1}^{k-1} \mathcal{P}_\alpha^{(i-1)} \left( \Delta W^{(i)} \right) \right) + \mathcal{P}_\alpha^{(k-1)} \left( \Delta W^{(k)} \right)}{\lambda^{(k)}}, \tag{17}$$

$$= W^{(0)} + \frac{1}{\lambda^{(k)}} \sum_{i=1}^{k} \mathcal{P}_\alpha^{(i-1)} \left( \Delta W^{(i)} \right). \tag{18}$$

By the principle of mathematical induction, the formula holds for all $t \geq 1$. $\qquad\square$

A fundamental property of our method is the orthogonality between the projected task vectors and the previously merged model. This orthogonality property is essential as it guarantees that new task vectors do not interfere with previously acquired knowledge, thereby preventing catastrophic forgetting.

*Proof of Theorem 5.1.* By definition of the projection mapping $\mathcal{P}_\alpha^{(t-1)}$, we have:

$$\left\langle \mathcal{P}_\alpha^{(t-1)}(\Delta W^{(t)}), \Delta W_{\text{merged}}^{(t-1)} \right\rangle_F = \left\langle \sum_{i,j=r_\alpha, i \neq j}^{m,n} \left\langle \Delta W^{(t)}, u_i v_j^T \right\rangle_F u_i v_j^T, U^{(t-1)} \Sigma^{(t-1)} V^{(t-1)T} \right\rangle_F, \tag{19}$$

$$= \sum_{i,j=r_\alpha, i \neq j}^{m,n} \left\langle \Delta W^{(t)}, u_i v_j^T \right\rangle_F \left\langle u_i v_j^T, U^{(t-1)} \Sigma^{(t-1)} V^{(t-1)T} \right\rangle_F. \tag{20}$$

Since $U^{(t-1)} \Sigma^{(t-1)} V^{(t-1)T}$ represents the SVD of $\Delta W_{\text{merged}}^{(t-1)}$, where the columns of $U^{(t-1)}$ and $V^{(t-1)}$ form orthonormal bases, for any pair $(i,j)$ where $i \neq j$, we have $\left\langle u_i v_j^T, U^{(t-1)} \Sigma^{(t-1)} V^{(t-1)T} \right\rangle_F = 0$ due to the orthogonality of singular vectors. Therefore:

$$\left\langle \mathcal{P}_\alpha^{(t-1)}(\Delta W^{(t)}), \Delta W_{\text{merged}}^{(t-1)} \right\rangle_F = 0. \tag{21}$$

$\qquad\square$

Finally, we prove the bounded distance theorem, which establishes that the distance between the merged model and the base model is bounded by the maximum distance between any task-specific model and the base model. This property provides a theoretical guarantee that the merged model remains within a controlled region around the base model, preventing catastrophic drift while allowing for effective adaptation to new tasks.

*Proof of Theorem 5.2.* We proceed by induction on $t$. For the base case $t = 1$, we have $W_{\text{merged}}^{(1)} - W^{(0)} = \Delta W^{(1)}$, so the bound holds trivially.

Assume the bound holds for $t - 1$. For step $t$, using Eq.(6) and the orthogonality property of the projection operator:

$$\left\| W_{\text{merged}}^{(t)} - W^{(0)} \right\|_F^2 = \left\| \frac{\sqrt{t-1} \Delta W_{\text{merged}}^{(t-1)} + \mathcal{P}_\alpha^{(t-1)}(\Delta W^{(t)})}{\sqrt{t}} \right\|_F^2, \tag{22}$$

$$= \frac{t-1}{t} \left\| \Delta W_{\text{merged}}^{(t-1)} \right\|_F^2 + \frac{1}{t} \left\| \mathcal{P}_\alpha^{(t-1)}(\Delta W^{(t)}) \right\|_F^2. \tag{23}$$

By the properties of the projection operator, we know that $\left\|\mathcal{P}_\alpha^{(t-1)}\left(\Delta W^{(t)}\right)\right\|_F^2 \leq \left\|\Delta W^{(t)}\right\|_F^2$. Applying the induction hypothesis:

$$\left\|W_{\text{merged}}^{(t)} - W^{(0)}\right\|_F^2 \leq \frac{t-1}{t} \max_{i \in [1,t-1]} \left\|\Delta W^{(i)}\right\|_F^2 + \frac{1}{t} \left\|\Delta W^{(t)}\right\|_F^2, \tag{24}$$

$$\leq \max_{i \in [1,t]} \left\|\Delta W^{(i)}\right\|_F^2. \tag{25}$$

This completes the induction, proving the bound holds for all $t \geq 1$. $\qquad\square$

## A.2 Theoretical Analysis of Correlated Task Vectors

Another important theoretical consideration in continual model merging is how the projection mechanism handles correlated task vectors, where two task-specific models are highly similar to each other. We analyze this scenario to demonstrate another advantageous property of our approach.

**Corollary A.2** (Efficient Handling of Correlated Task Vectors). *Consider two task vectors $\Delta W^{(t-1)}$ and $\Delta W^{(t)}$ with high correlation. After incorporating $\Delta W^{(t-1)}$ into $W_{\text{merged}}^{(t-1)}$, the contribution of $\Delta W^{(t)}$ to the merged model is automatically filtered to include only novel information not present in $W_{\text{merged}}^{(t-1)}$.*

*Proof.* Due to the high correlation between task vectors, we can decompose $\Delta W^{(t)}$ as follows:

$$\Delta W^{(t)} = \rho \Delta W_{\text{merged}}^{(t-1)} + \eta, \tag{26}$$

where $\rho$ is a scalar representing the correlation coefficient (high when vectors are strongly correlated) and $\eta$ is a component orthogonal to $\Delta W_{\text{merged}}^{(t-1)}$, i.e., $\langle \eta, \Delta W_{\text{merged}}^{(t-1)} \rangle_F = 0$. Applying our projection operator, we have:

$$\mathcal{P}_\alpha^{(t-1)}(\Delta W^{(t)}) = \mathcal{P}_\alpha^{(t-1)}(\rho \Delta W_{\text{merged}}^{(t-1)} + \eta) \tag{27}$$

$$= \rho \mathcal{P}_\alpha^{(t-1)}(\Delta W_{\text{merged}}^{(t-1)}) + \mathcal{P}_\alpha^{(t-1)}(\eta) \tag{28}$$

By Theorem 5.1, $\mathcal{P}_\alpha^{(t-1)}(\Delta W_{\text{merged}}^{(t-1)}) = 0$ since $\Delta W_{\text{merged}}^{(t-1)}$ is orthogonal to its own projection. Additionally, since $\eta$ is already orthogonal to $\Delta W_{\text{merged}}^{(t-1)}$, we have $\mathcal{P}_\alpha^{(t-1)}(\eta) = \eta$. Therefore:

$$\mathcal{P}_\alpha^{(t-1)}(\Delta W^{(t)}) = \eta \tag{29}$$

Consequently, the update for the merged model becomes:

$$W_{\text{merged}}^{(t)} = W^{(0)} + \frac{\lambda^{(t-1)} \Delta W_{\text{merged}}^{(t-1)} + \eta}{\lambda^{(t)}} \tag{30}$$

When the correlation is high, $|\eta|$ will be small, resulting in a minimal parameter update from the second task, which is precisely what we want for highly correlated task vectors. This demonstrates that the orthogonal projection mechanism of our method:

1. Automatically filters redundant knowledge in $\Delta W^{(t)}$ that is already captured in $W_{\text{merged}}^{(t-1)}$.

2. Incorporates only the novel information ($\eta$) into the merged model.

3. Prevents over-representation of similar concepts across multiple tasks, which could lead to catastrophic forgetting or overfitting to specific patterns.

$\qquad\square$

This result has important practical implications. In scenarios where new task models are highly similar to previously merged models, our projection approach automatically adjusts the magnitude of parameter updates based on the novelty of the information. This automatic filtering mechanism contributes to the stability and effectiveness of the continual merging process, particularly when dealing with a sequence of related tasks.

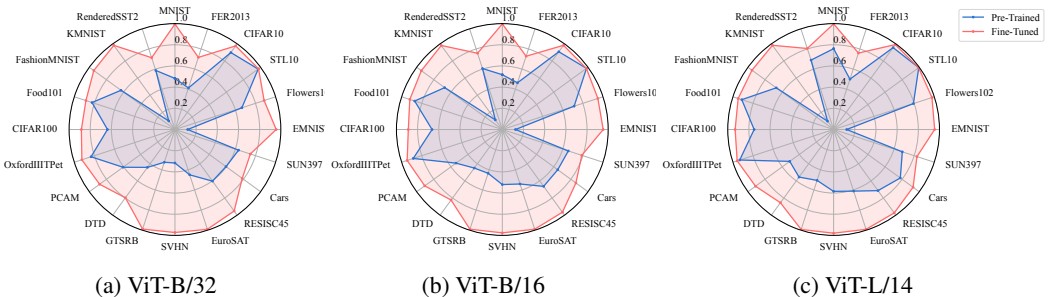

(a) ViT-B/32      (b) ViT-B/16      (c) ViT-L/14

Figure 8: Comparison of test set accuracy between the pre-trained model and individual fine-tuned models on different downstream tasks.

## B  Computational Complexity

We briefly discussed the memory complexity in Section 3.2, showing $O(|\theta|)$ constant memory complexity for continual model merging versus $O(T|\theta|)$ for conventional merging approaches. Here we discuss the computational complexity of OPCM. The computational complexity of OPCM is primarily dominated by the SVD decomposition, for each weight matrix $W \in \mathbb{R}^{m \times n}$, computing the full SVD has complexity $O(\min(m^2 n, n^2 m))$. Therefore, for $T$ tasks and $L$ linear layers, the complexity is approximately $O(TL \min(mn, mn))$.

Table 2: This table shows the runtime of different methods.

| Method | Wall-Clock Time of Merging | Wall-Clock Time per Model |
|---|---|---|
| Continual Task Arithmetic | 0.37s | $\approx$0.04s |
| Continual Ties Merging | 15.65s | $\approx$2s |
| AdaMerging (Conventional Model Merging) | 10.47min | $\approx$78.5s |
| AdaMerging (Continual Model Merging) | 11.85min | $\approx$88.9s |
| **OPCM** | 73.26s | $\approx$9.1s |

In Table 2, we show the runtime of different methods, including AdaMerging [96], Continual Task Arithmetic [29], Continual Ties Merging [91], and OPCM (the details of the compared baseline methods can be found in Appendices D.3.1 and E). As shown in the Table, our method is much faster than test-time adaptation-based methods like AdaMerging, while maintaining runtime in the same order of magnitude as training-free methods like Task Arithmetic and Ties-Merging.

## C  Details of the Fine-Tuned Models

In this section, we present the experimental details of model fine-tuning and evaluate the performance of the pre-trained model (zero-shot test accuracy) and our fine-tuned models on the test set of each downstream task. For each downstream task, we fine-tuned the visual encoder of the pre-trained CLIP-ViT models using task-specific training data, the classification heads are initialized using the pre-trained text encoder and fixed throughout the fine-tuning process. We employed a standard fine-tuning protocol with cross-entropy loss and the Adam optimizer, using a cosine annealing learning rate schedule with a maximum learning rate of 1e-5 and batch size 128, and train for 4000 steps [2].

**Models and datasets**. We adopt the same experimental setup as described in [83], where we fine-tune three different sizes of CLIP-ViT models ( ViT-B/32, ViT-B/16, and ViT-L/14 ) on 20 downstream image classification tasks. All the models and datasets are publicly available on Hugging Face [85]. The 20 downstream tasks are SUN397 [88], Cars [39], RESISC45 [10], EuroSAT [26], SVHN [55], GTSRB [71], MNIST [41], DTD [11], Flowers102 [57], PCAM [80], FER2013 [24], OxfordIIITPet [60], STL10 [13], CIFAR100, CIFAR10 [40], Food101 [4], FashionMNIST [87], EMNIST [14], KMNIST [12], and RenderedSST2 [70, 63].

---

[2] The models are fine-tuned and evaluated using the FusionBench [75].

Table 3: Test set accuracy of the pre-trained model and individual fine-tuned models on different downstream tasks.

| Model | SUN397 | Cars | RESISC45 | EuroSAT | SVHN | GTSRB | MNIST | DTD | Flowers102 | PCAM |
|---|---|---|---|---|---|---|---|---|---|---|
| | | | | *CLIP-ViT-B/32* | | | | | | |
| Pre-trained | 63.2 | 59.6 | 60.3 | 45.0 | 31.6 | 32.5 | 48.3 | 44.2 | 66.4 | 60.6 |
| Fine-tuned | 74.9 | 78.5 | 95.1 | 99.1 | 97.3 | 98.9 | 99.6 | 79.7 | 88.6 | 88.0 |
| | | | | *CLIP-ViT-B/16* | | | | | | |
| Pre-trained | 65.5 | 64.7 | 66.4 | 54.1 | 52.0 | 43.5 | 51.7 | 45.0 | 71.3 | 54.0 |
| Fine-tuned | 78.9 | 85.9 | 96.6 | 99.0 | 97.6 | 99.0 | 99.7 | 82.3 | 94.9 | 90.6 |
| | | | | *CLIP-ViT-L/14* | | | | | | |
| Pre-trained | 68.2 | 77.9 | 71.3 | 61.2 | 58.4 | 50.5 | 76.3 | 55.5 | 79.2 | 51.2 |
| Fine-tuned | 82.8 | 92.8 | 97.4 | 99.1 | 97.9 | 99.2 | 99.8 | 85.5 | 97.7 | 91.1 |

| Model | FER2013 | OxfordIIITPet | STL10 | CIFAR100 | CIFAR10 | Food101 | FashionMNIST | EMNIST | KMNIST | RenderedSST2 |
|---|---|---|---|---|---|---|---|---|---|---|
| | | | | *CLIP-ViT-B/32* | | | | | | |
| Pre-trained | 41.3 | 83.3 | 97.1 | 63.7 | 89.8 | 82.4 | 63.0 | 12.0 | 10.0 | 58.6 |
| Fine-tuned | 71.6 | 92.5 | 97.5 | 88.4 | 97.6 | 88.4 | 94.7 | 95.6 | 98.2 | 71.3 |
| | | | | *CLIP-ViT-B/16* | | | | | | |
| Pre-trained | 46.4 | 88.4 | 98.3 | 66.3 | 90.8 | 87.0 | 67.3 | 12.4 | 11.2 | 60.6 |
| Fine-tuned | 72.8 | 94.5 | 98.2 | 88.8 | 98.3 | 91.9 | 94.5 | 95.3 | 98.1 | 75.7 |
| | | | | *CLIP-ViT-L/14* | | | | | | |
| Pre-trained | 50.0 | 93.2 | 99.4 | 75.1 | 95.6 | 91.2 | 67.0 | 12.3 | 9.7 | 68.9 |
| Fine-tuned | 75.9 | 95.7 | 99.2 | 93.0 | 99.1 | 94.8 | 95.3 | 95.4 | 98.3 | 80.5 |

In the fine-tuning process, only the vision encoder was adjusted, while the text encoder remained unchanged. The pre-trained text encoder was used to generate (zero-shot)task-specific classification heads, and the classification heads were also fixed throughout the fine-tuning process. This means that both the pre-trained and fine-tuned models utilize the same classification heads for a given task. This approach preserves the open-vocabulary capability of the model, allowing it to remain applicable to any downstream task. Additionally, freezing the classification head does not compromise accuracy as shown in [28].

Figure 8 and Table 3 demonstrate the performance comparison between pre-trained and fine-tuned models across 20 downstream tasks. As illustrated, fine-tuning consistently improves model performance across all architectures (ViT-B/16, ViT-B/32, and ViT-L/14), with particularly notable gains in task-specific accuracy for most tasks.

Figures 9, 10, and 11 present the cosine similarity matrices between task vectors for CLIP-ViT models of different sizes (B/32, B/16, and L/14) fine-tuned on 20 downstream tasks. Several key observations emerge from these visualizations: (1) Most off-diagonal elements show very low cosine similarity values (0.01-0.05), indicating that the majority of task vectors are largely orthogonal to each other. (2) The similarity patterns remain largely consistent across different model architectures, though ViT-L/14 shows slightly lower similarity values overall, suggesting that larger models may learn more specialized representations for each task. (3) Some groups of related tasks show slightly elevated similarity values, indicating that the tasks are more similar to each other. For example,

- Digit recognition tasks (SVHN,MNIST, EMNIST, KMNIST) exhibit higher mutual similarities.
- Natural image classification tasks (CIFAR10, CIFAR100) show moderate correlations.

In most of the experiments, we group the tasks into 8, 14, and 20 tasks, and report the average accuracy (ACC) and backward transfer (BWT) of the merged models. The task groups are as follows:

- The 8 tasks: SUN397, Cars, RESISC45, EuroSAT, SVHN, GTSRB, MNIST, and DTD
- The 14 tasks: the 8 tasks together with Flowers102, PCAM, FER2013, OxfordIIITPet, STL10, and CIFAR100.
- The 20 tasks: the 14 tasks together with CIFAR10, Food101, FashionMNIST, EMNIST, KMNIST, and RenderedSST2.

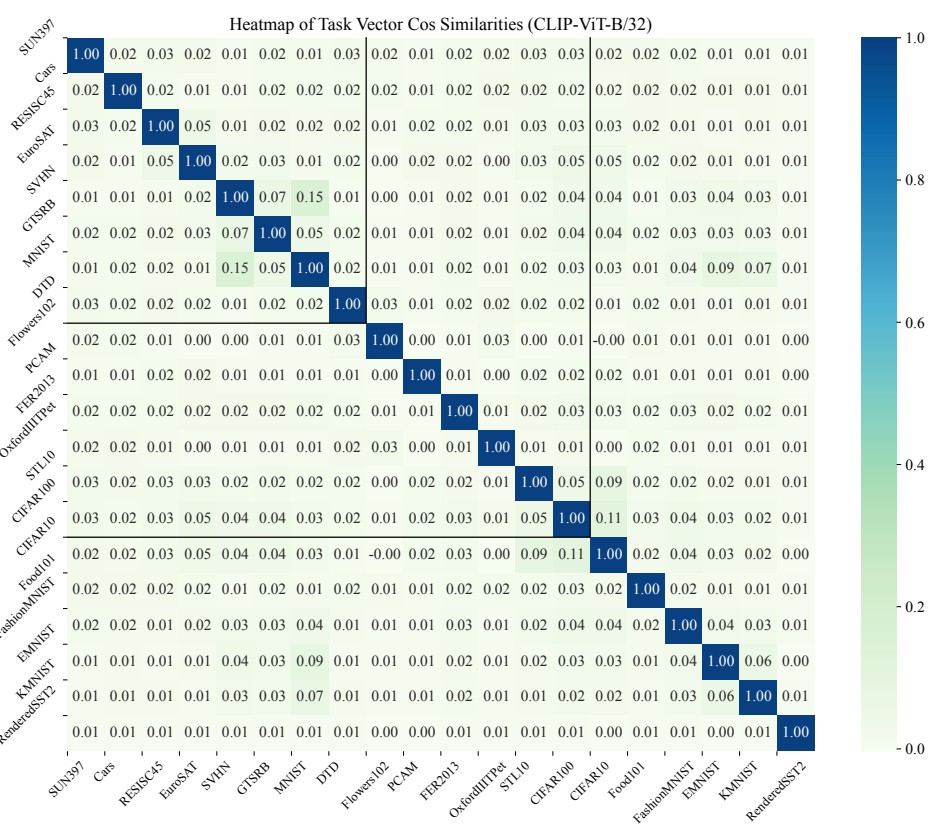

Figure 9: Cosine similarity between task vectors of CLIP-ViT-B/32 models fine-tuned on different downstream tasks.

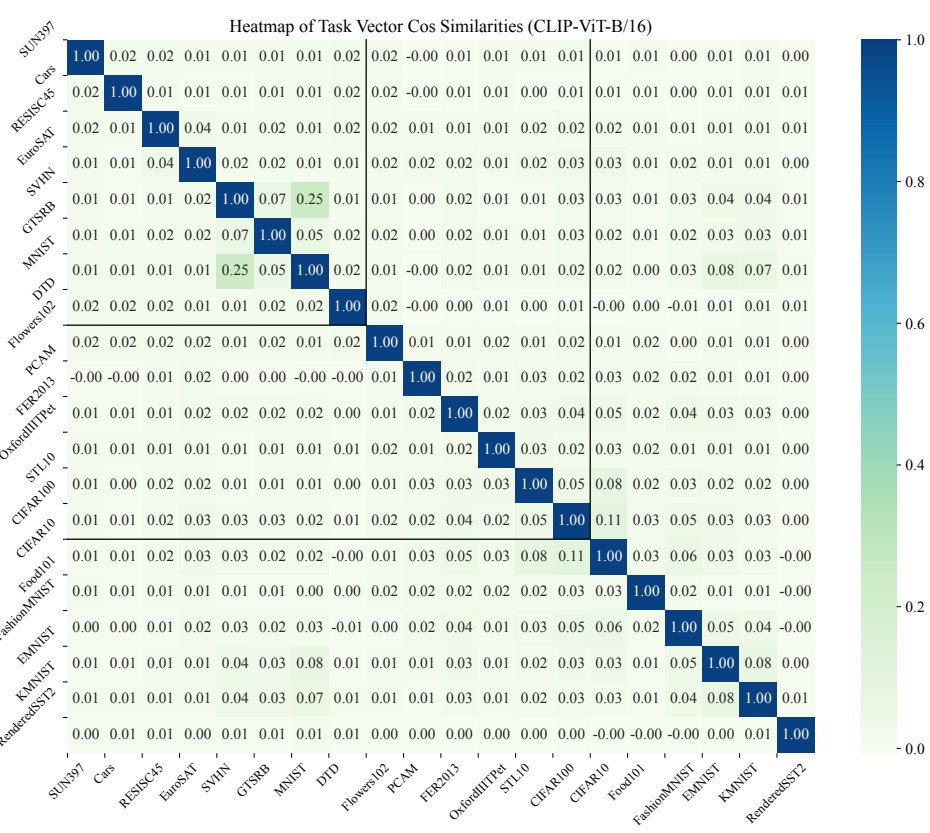

Figure 10: Cosine similarity between task vectors of CLIP-ViT-B/16 models fine-tuned on different downstream tasks.

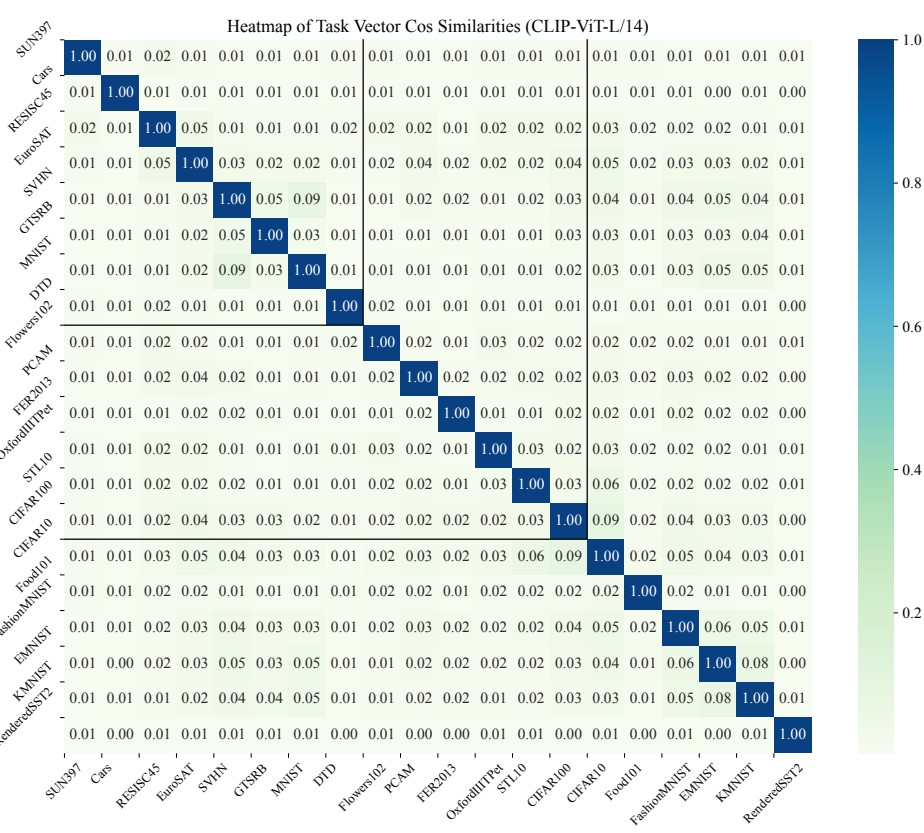

Figure 11: Cosine similarity between task vectors of CLIP-ViT-L/14 models fine-tuned on different downstream tasks.

# D  Additional Details on Experiments

## D.1  Computational Resources

All experiments were conducted on a single machine with 8 NVIDIA RTX 4090 GPUs, each with 24GB of memory. The code was implemented in PyTorch and Python 3.12.

## D.2  Evaluation Metrics

Here we provide the details of the evaluation metrics used in our experiments, including average accuracy (ACC) of the merged model at the final step and backward transfer (BWT).

$$\text{ACC}_{\text{avg}} = \frac{1}{T} \sum_{t=1}^{T} \text{ACC}_i \left( W_{\text{merged}}^{(T)} \right), \tag{31}$$

$$\text{BWT} = \frac{\sum_{i=1}^{T-1} \text{ACC}_i \left( \theta_{\text{merged}}^{(T)} \right) - \text{ACC}_i \left( \theta_{\text{merged}}^{(i)} \right)}{T-1}, \tag{32}$$

where $\theta_{\text{merged}}^{(i)}$ and $\theta_{\text{merged}}^{(T)}$ are the parameters of the merged model at $i$-th step and the final merged model, respectively. $\text{ACC}_i(\cdot)$ is the accuracy on the $i$-th task's test data.

## D.3  Continual Multi-Task Model Merging

For our continual multi-task model merging experiments, we evaluated three different CLIP-ViT architectures (ViT-B/32, ViT-B/16, and ViT-L/14) on task sets of increasing size (8, 14, and 20 tasks). Each experiment was repeated 10 times with randomly shuffled task orders to ensure robust evaluation. The models were initialized using pre-trained CLIP checkpoints and subsequently fine-tuned separately for each task before being merged. During fine-tuning, only the vision encoder was updated, while the text encoder remained frozen and was employed to generate task-specific classification heads. For further details on the fine-tuning process, please see Appendix C.

### D.3.1  Baseline Methods

In our experiments, we follow the common practice in model merging literature of using zero-shot classification heads initialized with the pre-trained CLIP model's text encoder. Specifically, we construct task-specific classification heads using CLIP's text encoder to generate zero-shot weights for each task's class names and templates. When fine-tuning the vision encoder, the classification head remains fixed. This is meant to preserve the open-vocabulary nature of the CLIP model, so it can still handle unseen classes during inference.

**Continual Fine-tuning** is a common baseline method for continual learning. For a sequence of tasks associate with training datasets $\{\mathcal{D}_1, \mathcal{D}_2, \ldots, \mathcal{D}_t\}$, the model is fine-tuned on each task sequentially. Denote the model parameters after fine-tuning on the $i$-th task as $\theta^{(i)}$. The update rule of continual fine-tuning is given by:

$$\theta^{(t)} = \text{Fine-tune}(\theta^{(t-1)}, \mathcal{D}_t), \tag{33}$$

where Fine-tune is the fine-tuning process. For each task, we use the same cosine annealing learning rate schedule with a maximum learning rate of 1e-5 and batch size 128, and train for 4000 steps. For continual fine-tuning, we set the random seed to 42 and shuffle the task order.

**Stochastic weight averaging (SWA)** is a common technique to stabilize the training process and improve generalization in model training [30]. The update rule of SWA is given by:

$$\theta_{\text{SWA}}^{(1)} = \theta^{(1)}, \quad \theta_{\text{SWA}}^{(t)} = \frac{\theta_{\text{SWA}}^{(t-1)}(t-1) + \theta^{(t)}}{t}, \tag{34}$$

where $\theta^{(t)}$ is the model parameters at step $t$, and $\theta_{\text{SWA}}^{(t)}$ is the averaged model parameters at step $t$. Expanding the above equation, we have:

$$\theta_{\text{SWA}}^{(t)} = \frac{1}{t} \sum_{i=1}^{t} \theta^{(i)}. \tag{35}$$

Therefore, this method is equivalent to averaging the parameters of all the checkpoints, which is also known as modelsoups in the literature [86, 107, 62].

**Task Arithmetic** is a simple yet powerful technique for merging multiple task-specific models into a unified multi-task model, as demonstrated in recent studies [29, 59]. This method linearly combines the parameters of task-specific fine-tuned models with those of the shared pre-trained base model, allowing the merged model to handle multiple tasks effectively at the same time. Mathematically, task arithmetic can be formalized as follows. Given a pre-trained model with parameters $\theta^{(0)}$ and a set of $T$ task-specific models fine-tuned from the pre-trained model with parameters $\theta^{(i)}$ for $i = 1, 2, \ldots, T$, the parameters of the merged model $\theta_{\text{merged}}$ are computed as:

$$\theta_{\text{merged}} = \theta^{(0)} + \lambda \sum_{i=1}^{T} \left( \theta^{(i)} - \theta^{(0)} \right), \tag{36}$$

where $\lambda$ is a scaling factor that is usually selected on a validation set using a grid search. In our experiments, we choose the value of $\lambda$ that achieves the highest average accuracy from the set $\{0.1, 0.2, 0.3, 0.4, 0.5, 0.6, 0.7, 0.8, 0.9, 1.0\}$. The experimental results show that $\lambda = 0.3$ is the optimal value for eight image classification tasks, and $\lambda = 0.1$ is the optimal value for 14 and 20 tasks.

In our continual multi-task model merging experiments, we use the following update rule of the merged model at step $t$:

$$\theta_{\text{merged}}^{(0)} = \theta^{(0)}, \quad \theta_{\text{merged}}^{(t)} = \theta_{\text{merged}}^{(t-1)} + \lambda \left( \theta^{(t)} - \theta^{(0)} \right), \tag{37}$$

where $\theta^{(t)}$ is the model parameters at step $t$, and $\theta_{\text{merged}}^{(t)}$ is the merged model parameters at step $t$.

*Memory complexity of task arithmetic*: As explained in Section 3.2, a continual model merging algorithm generally requires $O(|\theta|)$ memory, where $|\theta|$ represents the size of a single model. And task arithmetic can be implemented as a continual model merging method naturally. However, the most naive implementation of task arithmetic under the conventional model merging setting requires $O(T|\theta|)$ memory since it loads all the task-specific models and obtain the merged model as $\theta_{\text{merged}} = \theta^{(0)} + \lambda \sum_{i=1}^{T} \left( \theta^{(i)} - \theta^{(0)} \right)$.

**Ties-Merging** further extends the task arithmetic method by incorporating a systematic approach to handle parameter redundancy and sign conflicts during model merging [91]. This method ensures that the merged model retains the most relevant and consistent information from the individual task-specific models, thereby improving overall performance and robustness. By trimming redundant parameters, electing the most significant sign for each parameter, and performing a disjoint merge, Ties-Merging effectively reduces interference between tasks, leading to more stable and accurate results. Mathematically, Ties-Merging can be formalized as follows:

$$\left\{ \tau_{\text{Ties}}^{(i)} \right\}_{i=1}^{T} = \text{TiesMerging} \left( \left\{ \tau^{(i)} \right\}_{i=1}^{T} \right), \tag{38}$$

$$\theta_{\text{merged}} = \theta^{(0)} + \lambda \sum_{i=1}^{T} \tau_{\text{Ties}}^{(i)}, \tag{39}$$

where $\theta^{(0)}$ is the pre-trained model parameters, $\tau^{(i)} = \theta^{(i)} - \theta^{(0)}$ is the task vector for the $i$-th task, and $\tau_{\text{Ties}}^{(i)}$ is the trimmed and selected task vector for the $i$-th task. $\lambda$ is a scaling factor that is similar to the one used in Task Arithmetic. We use the same scaling factor $\lambda$ as in Task Arithmetic.

In our continual multi-task model merging experiments, we use the following update rule of the merged model at step $t$:

$$\theta_{\text{merged}}^{(1)} = \theta^{(0)} + \lambda(\theta^{(1)} - \theta^{(0)}), \quad \theta_{\text{merged}}^{(t)} = \theta^{(0)} + \lambda \left( \tau_{\text{Ties-merged}}^{(t-1)} + \tau_{\text{Ties}}^{(t)} \right), \tag{40}$$

where $\left\{ \tau_{\text{Ties-merged}}^{(t-1)}, \theta_{\text{Ties}}^{(t)} \right\} = \text{TiesMerging} \left( \left\{ \theta_{\text{merged}}^{(t-1)} - \theta^{(0)}, \theta^{(t)} - \theta^{(0)} \right\} \right)$.

### D.3.2 Experimental Results

We conduct the continual multi-task model merging experiments on the three task groups using different sizes of CLIP-ViT models (ViT-B/32, ViT-B/16, and ViT-L/14). For each task group and

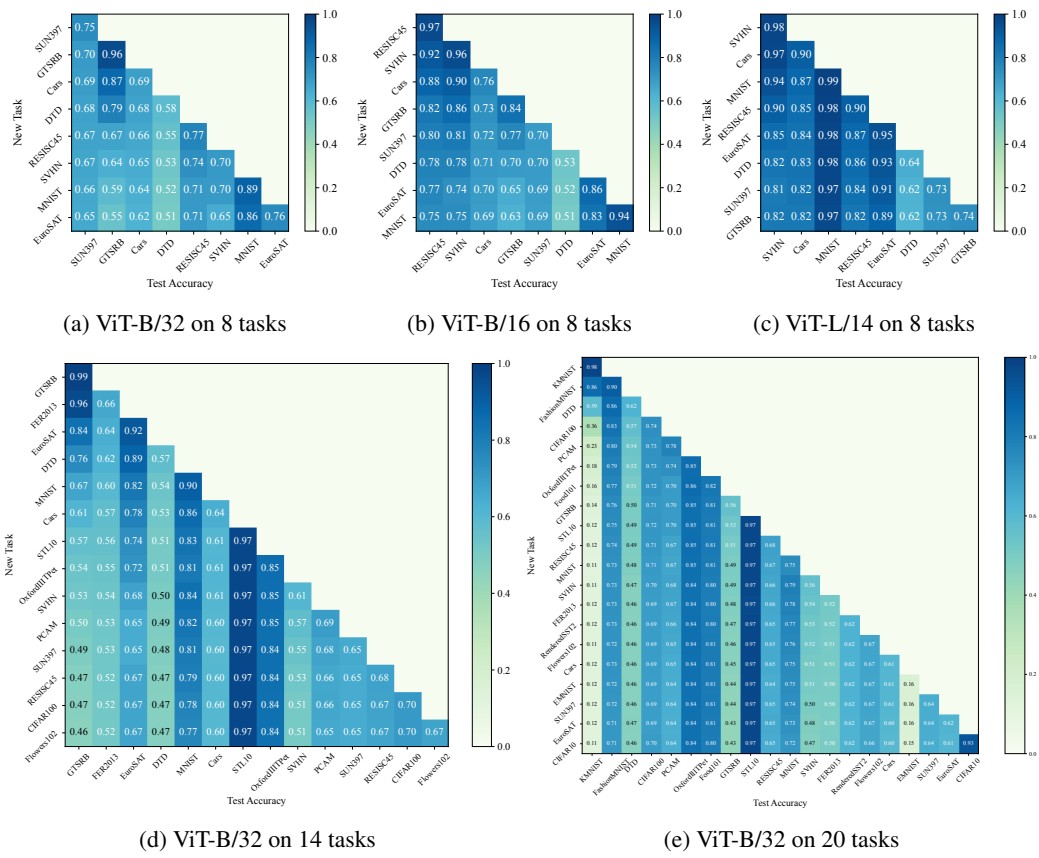

(a) ViT-B/32 on 8 tasks  (b) ViT-B/16 on 8 tasks  (c) ViT-L/14 on 8 tasks

(d) ViT-B/32 on 14 tasks  (e) ViT-B/32 on 20 tasks

Figure 12: Here we show the accuracy matrix of the continual multi-task model merging using weight averaging for different model architectures and task sets. The order of the tasks is shuffled for each run.

model size, we repeat the experiment 10 times with different task orders to investigate the effect of task order on the performance of the continual multi-task model merging.

**Accuracy matrix analysis.** The accuracy matrix is structured as follows: each row corresponds to a new task-specific model incorporated into the merged model, while each column displays the test accuracy for individual tasks after merging. Specifically, for the $i$-th row, the $j$-th column represents the test accuracy of the $j$-th task for the merged model $\theta_{\text{merged}}^{(i)}$, i.e. $\text{ACC}_j(\theta_{\text{merged}}^{(i)})$. The diagonal elements represent the performance on the newly added task, while off-diagonal elements show how well the model maintains performance on previously learned tasks.

In Figure 12, we show the accuracy matrix of the continual multi-task model merging using weight averaging for different model architectures and task sets. These accuracy matrices reveals several key insights about the continual multi-task model merging process. First, we observe a general trend of accuracy degradation as more tasks are added. This suggests some degree of catastrophic forgetting occurs during continual merging. Second, larger models (ViT-L/14) show better resistance to this degradation compared to smaller models (ViT-B/32), maintaining higher accuracies across tasks. Third, when comparing task sets of varying sizes (8, 14, and 20 tasks), it becomes evident that maintaining consistent performance across all tasks grows increasingly challenging as the number of tasks expands. This is reflected in the generally lower accuracy values observed in the 14- and 20-task matrices. Finally, we note that the performance decline is particularly significant for certain tasks, such as KMNIST and EMNIST. This could be attributed to the fact that these tasks exhibit the greatest divergence from the pre-trained model and the other tasks in the set.

**Per-task performance analysis.** The experimental results in Table 4 and Table 5 demonstrate the effectiveness of different model merging approaches across three CLIP-ViT architectures (B/32,

Table 4: Test set accuracy of the pre-trained model and individual fine-tuned models on 14 different downstream tasks. For continual fine-tuning, we fix the random seed to 42, shuffle the task order, and report the average accuracy. For other methods, we repeat the experiment 10 times with different task orders. Here we abbreviate 'Continual' as 'C.' to save space.

| Method | SUN397 | Cars | RESISC45 | EuroSAT | SVHN | GTSRB | MNIST |
|---|---|---|---|---|---|---|---|
| *CLIP-ViT-B/32* | | | | | | | |
| C. Fine-Tuned | 56.9 | 57.0 | 56.9 | **79.2** | **89.7** | **81.0** | **93.4** |
| Average (SWA) | 64.8±0.0 | **60.4±0.0** | 67.1±0.0 | 67.0±0.0 | 50.7±0.0 | 45.6±0.0 | 76.6±0.0 |
| C. Task Arithmetic | 64.4±0.0 | 59.6±0.0 | 67.3±0.0 | 67.8±0.0 | 54.0±0.0 | 50.0±0.0 | 80.7±0.0 |
| C. Ties-Merging | 64.8±0.8 | 59.7±1.7 | 68.9±4.7 | 62.5±3.1 | 50.1±2.8 | 53.3±7.2 | 80.8±5.0 |
| **OPCM (Ours)** | **65.6±0.4** | 57.9±1.0 | **70.8±1.8** | 77.8±2.1 | 69.9±1.8 | 62.3±1.0 | 92.2±1.0 |
| *CLIP-ViT-B/16* | | | | | | | |
| C. Fine-Tuned | 64.3 | **69.9** | 66.2 | **86.6** | **93.3** | **79.6** | 94.9 |
| Average (SWA) | 67.5±0.0 | 65.9±0.0 | 71.5±0.0 | 71.1±0.0 | 64.6±0.0 | 54.1±0.0 | 82.6±0.0 |
| C. Task Arithmetic | 67.8±0.0 | 65.2±0.0 | 71.9±0.0 | 74.0±0.0 | 67.0±0.0 | 57.2±0.0 | 88.9±0.0 |
| C. Ties-Merging | 68.5±0.7 | 67.2±2.7 | 72.3±1.5 | 73.0±7.1 | 65.6±5.0 | 53.8±3.0 | 88.6±4.1 |
| **OPCM (Ours)** | **68.8±0.3** | 64.4±1.8 | **78.7±1.2** | 84.6±1.9 | 79.2±1.9 | 70.5±2.5 | **95.4±0.5** |
| *CLIP-ViT-L/14* | | | | | | | |
| C. Fine-Tuned | 53.8 | 78.7 | 59.7 | 73.3 | **89.0** | **99.0** | **98.8** |
| Average (SWA) | 71.2±0.0 | 79.0±0.0 | 78.7±0.0 | 80.4±0.0 | 71.3±0.0 | 64.6±0.0 | 94.3±0.0 |
| C. Task Arithmetic | 71.6±0.0 | 78.4±0.0 | 79.3±0.0 | 80.3±0.0 | 72.4±0.0 | 67.9±0.0 | 95.3±0.0 |
| C. Ties-Merging | 72.2±1.2 | 78.4±0.8 | 81.5±4.9 | 79.4±5.3 | 72.4±4.0 | 66.1±2.8 | 94.6±1.5 |
| **OPCM (Ours)** | **74.0±0.2** | **82.1±0.4** | **85.8±0.8** | **86.2±1.3** | 84.4±1.8 | 85.1±1.3 | 97.7±0.2 |

| Method | DTD | Flowers102 | PCAM | Fer2013 | OxfordIIITPet | STL10 | CIFAR100 |
|---|---|---|---|---|---|---|---|
| *CLIP-ViT-B/32* | | | | | | | |
| C. Fine-Tuned | 47.4 | 38.8 | 69.4 | **70.4** | 72.6 | 87.8 | 43.7 |
| Average (SWA) | 46.9±0.0 | **67.4±0.0** | 65.2±0.0 | 51.6±0.0 | 84.2±0.0 | **97.2±0.0** | **70.4±0.0** |
| C. Task Arithmetic | 48.0±0.0 | 66.1±0.0 | 69.8±0.0 | 53.1±0.0 | 84.2±0.0 | 96.6±0.0 | 69.2±0.0 |
| C. Ties-Merging | 48.2±1.6 | 66.0±2.2 | 67.6±2.8 | 54.3±3.1 | 84.0±0.7 | 97.1±0.3 | 69.9±2.3 |
| **OPCM (Ours)** | **52.3±0.7** | 65.6±0.6 | **81.2±1.2** | 60.5±0.5 | **84.6±0.2** | 96.7±0.2 | 68.5±0.5 |
| *CLIP-ViT-B/16* | | | | | | | |
| C. Fine-Tuned | 46.4 | 42.0 | 64.1 | **72.7** | 84.1 | 90.8 | 56.5 |
| Average (SWA) | 47.2±0.0 | 72.5±0.0 | 63.2±0.0 | 54.1±0.0 | 90.4±0.0 | **98.3±0.0** | 73.4±0.0 |
| C. Task Arithmetic | 47.8±0.0 | 72.0±0.0 | 64.8±0.0 | 54.8±0.0 | 90.6±0.0 | 98.1±0.0 | 72.8±0.0 |
| C. Ties-Merging | 48.1±0.9 | 71.5±1.0 | 63.2±1.7 | 54.5±3.5 | 90.3±1.0 | 98.2±0.1 | 72.5±2.8 |
| **OPCM (Ours)** | **51.8±0.7** | **74.5±0.7** | **83.9±1.2** | 62.7±0.4 | **92.7±0.2** | 97.9±0.1 | **73.8±1.2** |
| *CLIP-ViT-L/14* | | | | | | | |
| C. Fine-Tuned | 42.6 | 57.3 | 60.9 | 52.9 | 83.1 | 87.6 | 57.1 |
| Average (SWA) | 58.7±0.0 | 81.9±0.0 | 74.2±0.0 | 54.8±0.0 | 94.6±0.0 | **99.3±0.0** | 82.4±0.0 |
| C. Task Arithmetic | 59.8±0.0 | 81.9±0.0 | 71.1±0.0 | 56.1±0.0 | 94.8±0.0 | 99.0±0.0 | 82.3±0.0 |
| C. Ties-Merging | 59.3±1.1 | 81.6±1.5 | 72.8±1.9 | 57.3±1.8 | 94.6±0.4 | 99.2±0.2 | 82.9±1.2 |
| **OPCM (Ours)** | **63.6±1.1** | **87.6±0.3** | **80.4±0.9** | **63.1±0.4** | **95.8±0.1** | 99.2±0.0 | **84.0±0.5** |

B/16, and L/14) on 14 and 20 downstream tasks, respectively. It is observed that OPCM (the proposed method) outperforms other continual model merging approaches in a majority of tasks. For instance, on ViT-B/32, OPCM achieves the best performance on several tasks including SUN397 (64.4%), RESISC45 (66.0%), SVHN (66.1%), GTSRB (56.0%), MNIST (90.2%), and PCAM (80.2%), FER2013 (58.5%), CIFAR10 (92.8%), and RenderedSST2 (64.6%). The results also reveal that certain tasks, particularly EMNIST and KMNIST, remain challenging for all continual model merging methods, though OPCM generally maintains better performance even on these difficult cases. Furthermore, Continual Ties-Merging shows significant higher standard deviations compared to OPCM across multiple tasks and model arichitectures, indicating that Continual Ties-Merging is much more sensitive to the order of tasks are merged, leading to inconsistent performance across different task orderings.

In Table 6, we dive deeper into the comparison between continual fine-tuning and OPCM by examining the average accuracy over 10 different task orders for OPCM. Continual fine-tuning is extremely computationally intensive. For instance, in our 20-task experiments, each task requires

Table 5: Test set accuracy of the pre-trained model and individual fine-tuned models on 20 different downstream tasks. For continual fine-tuning, we fix the random seed to 42, shuffle the task order, and report the average accuracy. For other methods, we repeat the experiment 10 times with different task orders. Here we abbreviate 'Continual' as 'C.' to save space.

| Method | SUN397 | Cars | RESISC45 | EuroSAT | SVHN | GTSRB | MNIST | DTD | Flowers102 | PCAM |
|---|---|---|---|---|---|---|---|---|---|---|
| *CLIP-ViT-B/32* | | | | | | | | | | |
| C. Fine-Tuned | 53.9 | 38.2 | 64.7 | **98.7** | 45.4 | 34.4 | 86.7 | **58.4** | 57.5 | 67.7 |
| Average (SWA) | 64.2±0.0 | **59.6**±0.0 | 64.8±0.0 | 60.9±0.0 | 47.3±0.0 | 43.1±0.0 | 71.8±0.00.0 | 46.4±0.0 | **66.5**±0.0 | 63.9±0.0 |
| C. Task Arithmetic | 62.0±0.0 | 53.7±0.0 | 60.9±0.0 | 58.1±0.0 | 48.5±0.0 | 48.9±0.0 | 79.4±0.0 | 46.1±0.0 | 61.1±0.0 | 73.4±0.0 |
| C. Ties-Merging | 62.5±2.2 | 49.1±4.3 | 55.8±2.7 | 50.9±5.2 | 54.6±13.4 | 49.3±6.4 | 82.0±5.8 | 46.7±3.7 | 58.5±3.7 | 69.9±4.1 |
| **OPCM (Ours)** | **64.4**±0.3 | 51.1±1.6 | **66.0**±1.2 | 71.7±2.1 | **66.1**±2.6 | **56.0**±1.2 | **90.2**±0.6 | 40.4±0.7 | 64.9±0.4 | **80.2**±0.6 |
| *CLIP-ViT-B/16* | | | | | | | | | | |
| C. Fine-Tuned | 62.7 | 58.0 | 67.6 | **99.1** | 46.0 | 29.2 | 93.9 | **61.9** | 64.1 | 75.2 |
| Average (SWA) | 67.1±0.0 | **64.6**±0.0 | 69.3±0.0 | 63.4±0.0 | 62.4±0.0 | 52.0±0.0 | 80.7±0.0 | 46.6±0.0 | 71.8±0.0 | 63.1±0.0 |
| C. Task Arithmetic | 65.8±0.0 | 57.5±0.0 | 63.8±0.0 | 59.5±0.0 | 64.7±0.0 | 54.0±0.0 | 88.8±0.0 | 45.3±0.0 | 67.5±0.0 | 67.1±0.0 |
| C. Ties-Merging | 64.2±1.6 | 52.9±7.8 | 60.9±5.2 | 53.0±8.3 | 62.8±10.1 | 48.8±5.0 | 88.4±4.2 | 45.0±2.5 | 61.3±3.7 | 68.5±6.4 |
| **OPCM (Ours)** | **67.9**±0.2 | 55.9±1.3 | **73.7**±0.8 | 77.5±2.9 | **74.4**±0.9 | **63.2**±1.9 | **94.1**±0.5 | 49.2±0.4 | **72.3**±0.3 | **79.6**±3.0 |
| *CLIP-ViT-L/14* | | | | | | | | | | |
| C. Fine-Tuned | 69.5 | 73.6 | 78.3 | **99.2** | 59.3 | 49.3 | 98.6 | 69.7 | 83.2 | **78.3** |
| Average (SWA) | 70.7±0.0 | 77.7±0.0 | 76.4±0.0 | 75.3±0.0 | 69.5±0.0 | 62.1±0.0 | 93.7±0.0 | 57.7±0.0 | 80.0±0.0 | 73.6±0.0 |
| C. Task Arithmetic | 70.4±0.0 | 74.1±0.0 | 73.9±0.0 | 66.3±0.0 | 69.9±0.0 | 65.6±0.0 | 95.1±0.0 | 56.6±0.0 | 78.6±0.0 | 70.4±0.0 |
| C. Ties-Merging | 69.6±1.5 | 70.3±3.5 | 65.3±2.9 | 47.9±7.5 | 76.1±8.8 | 63.6±7.1 | 94.7±0.9 | 54.4±2.1 | 77.9±2.7 | 72.3±3.0 |
| **OPCM (Ours)** | **73.1**±0.2 | **78.3**±0.5 | **82.4**±0.4 | 80.2±2.6 | **80.8**±0.4 | **80.4**±0.7 | 97.4±0.3 | 61.6±0.8 | **84.8**±0.4 | 76.3±1.9 |

| Method | FER2013 | OxfordIIITPet | STL10 | CIFAR100 | CIFAR10 | Food101 | FashionMNIST | EMNIST | KMNIST | RenderedSST2 |
|---|---|---|---|---|---|---|---|---|---|---|
| *CLIP-ViT-B/32* | | | | | | | | | | |
| C. Fine-Tuned | 58.3 | 68.5 | 86.7 | 40.2 | 70.5 | 50.0 | **90.7** | **72.4** | **54.5** | 54.5 |
| Average (SWA) | 50.2±0.0 | **84.1**±0.0 | **97.0**±0.0 | **69.8**±0.0 | 92.7±0.0 | **80.4**±0.0 | 71.3±0.0 | 15.0±0.0 | 11.5±0.0 | 61.8±0.0 |
| C. Task Arithmetic | 51.4±0.0 | 82.3±0.0 | 94.9±0.0 | 64.6±0.0 | 91.4±0.0 | 71.9±0.0 | 73.9±0.0 | 17.8±0.0 | 12.2±0.0 | 59.9±0.0 |
| C. Ties-Merging | 49.5±2.9 | 81.3±1.9 | 95.2±0.5 | 63.7±1.8 | 91.2±1.4 | 70.2±2.6 | 73.7±4.5 | 17.8±2.8 | 16.9±4.6 | 59.8±4.1 |
| **OPCM (Ours)** | **58.5**±0.6 | 82.9±0.3 | 95.9±0.3 | 67.6±0.7 | **92.8**±0.3 | 74.0±0.8 | 76.3±1.0 | 22.4±1.0 | 18.3±1.6 | **64.6**±0.6 |
| *CLIP-ViT-B/16* | | | | | | | | | | |
| C. Fine-Tuned | **60.5** | 84.5 | 90.5 | 38.8 | 73.6 | 61.9 | **89.7** | **83.3** | 51.5 | **72.8** |
| Average (SWA) | 50.9±0.0 | 89.6±0.0 | **98.0** | 72.9±0.0 | 94.2±0.0 | **85.9**±0.0 | 73.3±0.0 | 15.6±0.0 | 12.4±0.0 | 62.5±0.0 |
| C. Task Arithmetic | 50.7±0.0 | 89.3±0.0 | 97.0±0.0 | 68.0±0.0 | 93.1±0.0 | 80.3±0.0 | 75.7±0.0 | 18.1±0.0 | 16.7±0.0 | 61.8±0.0 |
| C. Ties-Merging | 50.4±6.2 | 87.9±1.6 | 96.3±0.9 | 63.1±4.8 | 91.7±1.9 | 78.0±2.3 | 75.0±4.0 | 23.4±8.4 | 24.9±7.1 | 61.5±4.9 |
| **OPCM (Ours)** | 59.5±0.4 | **91.8**±0.2 | 97.7±0.1 | **73.2**±0.9 | **94.7**±0.2 | 83.1±0.2 | 81.3±0.6 | 26.5±1.2 | 23.4±0.6 | 66.8±0.8 |
| *CLIP-ViT-L/14* | | | | | | | | | | |
| C. Fine-Tuned | **68.0** | 92.1 | 94.5 | 60.5 | 85.7 | 74.8 | **93.1** | 89.0 | 59.2 | **78.8** |
| Average (SWA) | 52.7±0.0 | 94.2±0.0 | **99.2**±0.0 | 81.7±0.0 | 97.0±0.0 | 90.7±0.0 | 77.4±0.0 | 16.1±0.0 | 10.4±0.0 | 66.1±0.0 |
| C. Task Arithmetic | 55.7±0.0 | 94.2±0.0 | 98.6±0.0 | 79.1±0.0 | 96.6±0.0 | 87.6±0.0 | 80.8±0.0 | 17.6±0.0 | 10.6±0.0 | 63.6±0.0 |
| C. Ties-Merging | 57.6±3.0 | 93.5±0.8 | 97.8±0.5 | 74.0±4.3 | 95.6±1.7 | 84.7±2.5 | 79.7±2.1 | 20.2±5.5 | 12.6±4.2 | 58.4±2.4 |
| **OPCM (Ours)** | 61.8±0.2 | **95.4**±0.1 | **99.2**±0.0 | **83.0**±0.4 | **97.8**±0.1 | **90.9**±0.2 | 86.0±0.2 | 26.4±0.9 | 14.7±2.2 | 71.0±1.1 |

Table 6: Test set accuracy comparison between continual fine-tuning and OPCM across different sizes of CLIP-ViT models and task sets.

| Experimental Setup | Continual Fine-tuning | OPCM (Ours, 10 runs) |
|---|---|---|
| ViT-B/32, 8 tasks | 79.8 | 75.5±0.5, Min: 74.8, Max:76.5 |
| ViT-B/16, 8 tasks | 82.9 | 81.8±0.3, Min: 81.3, Max: 82.3 |
| ViT-L/14, 8 tasks | 90.0 | 87.0±0.4, Min: 86.6, Max: 87.7 |
| ViT-B/32, 14 tasks | 67.4 | 71.9±0.3, Min: **71.3**, Max: 72.5 |
| ViT-B/16, 14 tasks | 72.2 | 77.1±0.5, Min: **76.2**, Max: 77.8 |
| ViT-L/14, 14 tasks | 70.9 | 83.5±0.2, Min: **83.1**, Max: 83.8 |
| ViT-B/32, 20 tasks | 62.6 | 65.7±0.2, Min: **65.3**, Max: 66.0 |
| ViT-B/16, 20 tasks | 68.2 | 70.3±0.2, Min: **69.8**, Max: 70.7 |
| ViT-L/14, 20 tasks | 77.7 | 76.0±0.2, Min: 75.6, Max: 76.4 |

20-30 minutes of fine-tuning. A single complete run, therefore, takes 8-10 hours. Here we report continual fine-tuning results from single runs, the comparison with our method's statistical analysis reveals compelling insights: In 5 out of 9 experimental settings (highlighted in bold), even our method's minimum performance exceeds continual fine-tuning's single-run results. In the remaining cases, our method still achieves competitive results within 1-5% of continual fine-tuning. OPCM

achieves the performance with orders of magnitude lower computational cost, requiring only a few seconds versus 20-30 minutes per task for continual fine-tuning.

## D.4 Ablation Studies

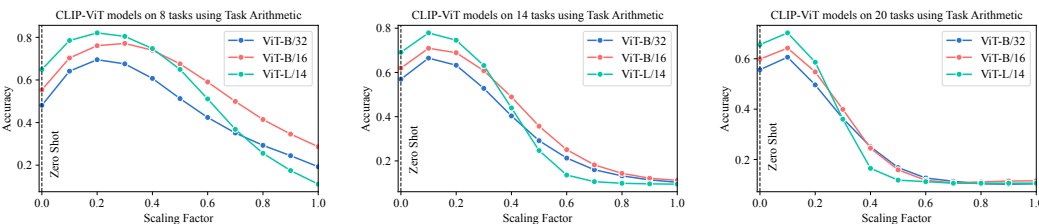

Figure 13: The average accuracy of Task Arithmetic with varying scaling factors.

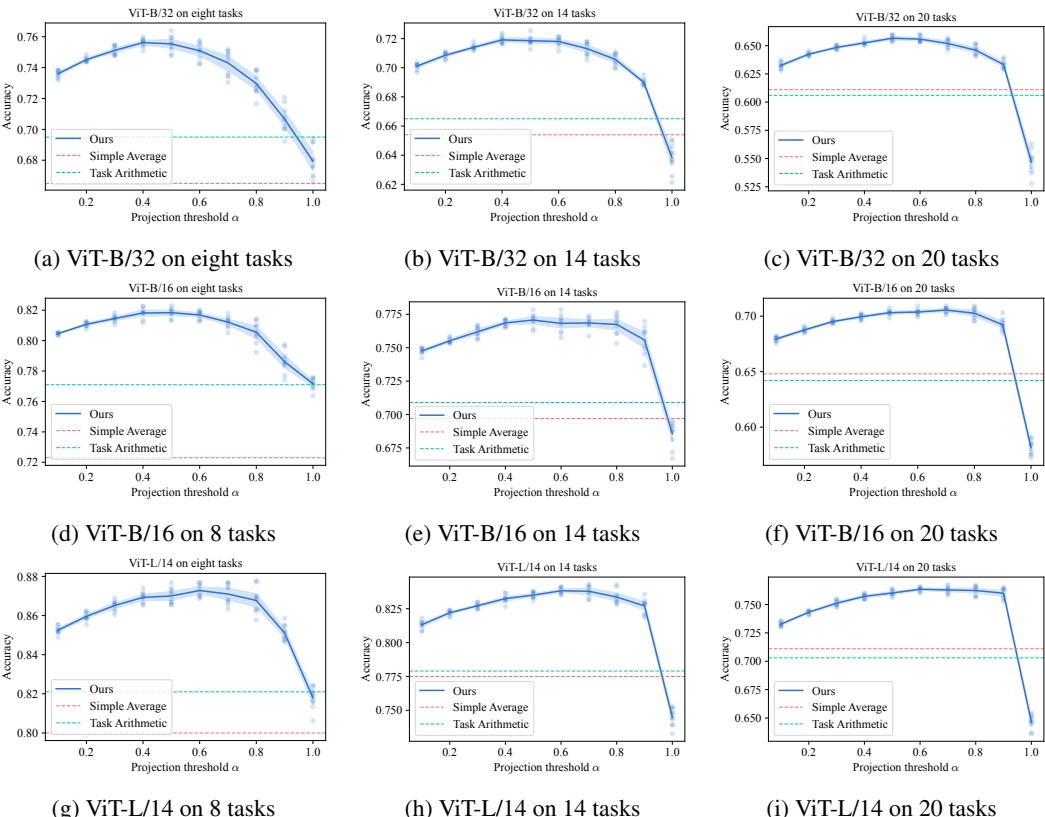

(a) ViT-B/32 on eight tasks    (b) ViT-B/32 on 14 tasks    (c) ViT-B/32 on 20 tasks

(d) ViT-B/16 on 8 tasks    (e) ViT-B/16 on 14 tasks    (f) ViT-B/16 on 20 tasks

(g) ViT-L/14 on 8 tasks    (h) ViT-L/14 on 14 tasks    (i) ViT-L/14 on 20 tasks

Figure 14: The effect of the projection threshold $\alpha$ on the performance of the merged CLIP-ViT models.

In this section, we provide additional details on the ablation studies. We perform a grid search to systematically evaluate the scaling factor for Task Arithmetic and the projection threshold $\alpha$ for our proposed method. This analysis is conducted across a range of model architectures, including ViT-B/32, ViT-B/16, and ViT-L/14, as well as diverse task sets comprising 8, 14, and 20 tasks. The hyperparameter search space is as follows:

- Scaling factor for Task Arithmetic: {0.0 (zero-shot), 0.1, 0.2, 0.3, 0.4, 0.5, 0.6, 0.7, 0.8, 0.9, 1.0}.
- Projection threshold $\alpha$ for our proposed method: {0.1, 0.2, 0.3, 0.4, 0.5, 0.6, 0.7, 0.8, 0.9, 1.0}.

Figure 13 demonstrates the impact of adjusting the scaling factor in Task Arithmetic across various model architectures (ViT-B/32, ViT-B/16, and ViT-L/14) and task sets (8, 14, and 20 tasks). The findings reveal that model performance is highly dependent on the choice of scaling factor, with distinct optimal values identified for different task sets. Specifically, a scaling factor of 0.3 yields the best results for eight tasks, while a factor of 0.1 proves optimal for both 14 and 20 tasks. These results highlight the importance of carefully selecting the scaling factor to maximize performance in multi-task learning scenarios.

Figure 14 shows the influence of the projection threshold $\alpha$ on the merged model performance, revealing several key insights. First, the optimal value of $\alpha$ consistently lies within the range of 0.4 to 0.6, regardless of the number of tasks. Second, performance remains remarkably stable within this range, highlighting the robustness of our method to slight variations in the choice of $\alpha$. Third, as the number of tasks increases from 8 to 20, the optimal range for $\alpha$ remains similar, demonstrating its scalability across different task complexities. Fourth, when compared to baseline methods such as Task Arithmetic, our approach consistently outperforms them across a broad spectrum of $\alpha$ values ($\alpha \leq 0.9$). These findings underscore the effectiveness of our method in facilitating efficient knowledge transfer and mitigating catastrophic forgetting, making it a reliable solution for continual multi-task model merging.

# E    Comparison With SOTA Conventional Model Merging Methods

Table 7: Comparison with conventional model merging methods on eight CLIP-ViT-B/32 models.

| Method | SUN397 | Cars | RESISC45 | EuroSAT | SVHN | GTSRB | MNIST | DTD | Avg. |
|---|---|---|---|---|---|---|---|---|---|
| *Conventional Model Merging* | | | | | | | | | |
| Fisher Merging | 66.7 | 64.0 | 72.2 | 91.6 | 69.0 | 64.3 | 83.5 | 53.7 | 70.6 |
| RegMean | 67.8 | 68.9 | 82.5 | 94.4 | 90.6 | 79.2 | 97.6 | 63.2 | 80.5 |
| Task Arithmetic | 57.1 | 55.7 | 64.9 | 76.7 | 77.9 | 68.5 | 96.1 | 47.2 | 68.0 |
| Ties-Merging | 67.1 | 64.2 | 74.1 | 76.8 | 77.7 | 69.4 | 94.1 | 54.0 | 72.2 |
| Task-wise AdaMerging | 58.6 | 56.9 | 69.8 | 82.4 | 70.3 | 58.9 | 97.2 | 55.3 | 68.7 |
| Layer-wise AdaMerging | 67.9 | 71.3 | 83.5 | 92.7 | 87.4 | 92.9 | 98.2 | 67.0 | 82.6 |
| TSVM | 67.6 | 71.6 | 84.7 | 93.4 | 91.9 | 92.5 | 98.9 | 63.8 | 83.1 |
| *Continual Model Merging* | | | | | | | | | |
| C. Layer-wise AdaMerging | 62.2 | 64.6 | 71.1 | 70.6 | 91.2 | 94.5 | 98.4 | 35.3 | 73.5 |
| C. TSVM | 59.9 | 59.5 | 64.0 | 64.7 | 87.9 | 86.1 | 99.5 | 34.3 | 68.5 |
| **Ours (10 runs)** | 66.2±0.5 | 63.5±0.5 | 78.0±2.2 | 86.5±1.9 | 82.4±1.3 | 74.3±1.4 | 96.9±0.7 | 56.4±1.3 | 75.5±0.5 |

In this study, we introduce the orthogonal projection-based continual merging (OPCM) method for scenarios where models *become available sequentially over time* rather than simultaneously. While conventional Model Merging such as AdaMerging assumes all expert models are available simultaneously. For a more comprehensive comparison, we also list the experiments results of some SOTA model merging methods on eight CLIP-ViT-B/32 models in Table 7. Our experimental results demonstrate that OPCM achieves superior performance compared to most conventional model merging approaches, despite operating under the more challenging continual setting where expert models are not simultaneously available, making it harder to resolve inter-task knowledge conflicts.

We also conducted experiments on AdaMerging [96] and TSVM [23] to compare its performance under the continual model merging setting. Specifically, we adapt the AdaMerging method and TSVM to the continual model merging setting by setting $\theta^{(0)}_{merged} = \theta^{(0)}$, $\theta^{(t+1)}_{merged} =$ AdaMerging$(\theta_0; \theta^{(t)}_{merged}, \theta^{(t+1)})$ and $\theta^{(t+1)}_{merged} = $ TSVM$(\theta_0; \theta^{(t)}_{merged}, \theta^{(t+1)})$, respectively. The results are shown in Table 7. The continual layer-wise AdaMerging and TSVM experiment was conducted with a single run using the following task sequence: SUN397, Cars, RESISC45, EuroSAT, SVHN, GTSRB, MNIST, and DTD. The results in the table demonstrate that AdaMerging in the continual merging setting achieves 73.5% average accuracy, which is notably lower than its performance in the conventional model merging setting (82.6%).

