# OpenReview forum: "Merging on the Fly Without Retraining: A Sequential Approach to Scalable Continual Model Merging"
_NeurIPS.cc/2025/Conference — NeurIPS 2025 poster_

### Official Review · Reviewer_Eirv · 2025-06-04

**Clarity:** 3
**Significance:** 2
**Originality:** 3
**Rating:** 4
**Confidence:** 4

**Summary:**

This paper proposes a novel continuous model merging method, named OPCM. Unlike traditional model merging approaches that require access to all models simultaneously, OPCM processes models one-by-one using orthogonal projections and adaptive scaling. The method maintains constant memory complexity and mitigates task interference. Extensive empirical evaluations using CLIP-ViT architectures on up to 20 image classification tasks show that OPCM outperforms strong baselines in both average accuracy and backward transfer (BWT), while remaining robust to task orderings.

**Questions:**

See weakness.

**Ethical Concerns:**

["NO or VERY MINOR ethics concerns only"]

**Final Justification:**

After the rebuttal, most of my concerns are addressed. Overall, I think this is a good paper, even though the proposed Continual Model Merging problem does not seem significant to the model merging community. However, considering that this paper is the first paper that proposed CMM, I would like to give a Weak Accept.

**Limitations:**

No. This paper uses the assumption that projecting the task vectors in the orthogonal space can cause less conflict to old tasks, which may not be valid for some scenarios.

**Paper Formatting Concerns:**

No formatting issue observed.

**Quality:**

3

**Strengths And Weaknesses:**

Strengths.
+ This paper is clearly written and easy to follow.
+ This paper introduces a new setting for model merging by incorporating the idea of continual learning.
+ Extensive experiments on various architectures and tasks show the effectiveness of the proposed method.

Weaknesses.
+ The paper proposes to solve the continuous model merging problem, where old task vectors cannot be stored. However, in practice, storing task vectors is usually not expensive, making this assumption somewhat questionable. So, I think this problem may not be a significant setting for model merging.
+ This paper uses the assumption that projecting the task vectors in the orthogonal space can cause less conflict with old tasks. Does this assumption always hold? Maybe some important dimension of old task vectors is zero just simply because the pretrained weights are good enough.

---

> ### Author Rebuttal · Authors · 2025-07-29
>
> Thank you for your thoughtful feedback. We’re grateful for the time and effort you’ve dedicated.
>
> ## Response to Weakness 1
>
> > The paper proposes to solve the continuous model merging problem, where old task vectors cannot be stored. However, in practice, storing task vectors is usually not expensive, making this assumption somewhat questionable. So, I think this problem may not be a significant setting for model merging.
>
> While disk storage is indeed available, the continual model merging setting is motivated by practical situations where models are frequently updated—often by different teams or individuals—and earlier versions may not be retained or accessible.
>
> Besides, the $O(T|\theta|)$ complexity isn't just about storage, it also affects the computational overhead during merging.
>
> ## Response to Weakness 2
>
> > This paper uses the assumption that projecting the task vectors in the orthogonal space can cause less conflict with old tasks. Does this assumption always hold? Maybe some important dimension of old task vectors is zero just simply because the pretrained weights are good enough.
>
> This is an excellent point that deserves deeper analysis. In our experiments across 20 tasks, we empirically observed that the orthogonal projection outperforms the continual ties-meriging method, which does not use orthogonal projection. This suggests that the orthogonal projection is indeed beneficial in reducing task interference.
>
> Besides conflict reduction, the projection operation also offers a significant advantage: it prevents knowledge redundancy. In a non-orthogonal approach, different tasks might encode similar features or information in their updates, duplicating effort and wasting the model’s capacity.
> For instance, if two tasks require recognizing similar low-level patterns, they might redundantly adjust parameters to represent those knowledge independently.
> In OPCM, the base model provides a foundation of shared features accessible to all tasks, and the orthogonal projection ensures that each task’s updates are unique and keep the merged model lie in a low-loss basin that is common to all tasks.
> By avoiding duplication, OPCM helps maintain the model’s generalization ability across tasks and prevents it from being pushed into a suboptimal region of the parameter space.
>
> > Maybe some important dimension of old task vectors is zero just simply because the pretrained weights are good enough.
>
> Thank you for raising this point.
> Recall that in OPCM we compute an SVD of the old cumulative update
> $$
> \Delta W_{\rm merged}^{(t-1)} = U\,Σ\,V^⊤,
> $$
> and then select only the top‑$r$ singular components whose values capture at least an α‑fraction of the total energy:
> $$
> \sum_{i=1}^r σ_i \;\ge\; α\;\sum_{j=1}^{\min(m,n)} σ_j.
> $$
> Any coordinate *k* for which $\Delta W_{\rm merged}^{(t-1)}$ has zero entry will contribute a singular value $σ_k=0$, and so its corresponding singular vectors $(u_k,v_k)$ lie outside our top‑$r$ subspace.
>
> Therefore, If an “important” dimension *k* is zero in $\Delta W_{\rm merged}^{(t-1)}$, then in practice that dimension corresponds to a zero singular value and is not selected into the top‑$r$ subspace.  Hence, our orthogonal projection does not eliminate $\Delta W^{(t)}$’s component along that axis.

---

> > ### Comment · Reviewer_Eirv · 2025-08-01
> >
> > Thank you for the clarification. However, I still find this motivation difficult to accept. In most practical scenarios, trained model parameters or at least lightweight task vectors are typically archived rather than discarded, precisely because training models is expensive and often the result of significant effort by teams. It is quite uncommon for earlier versions or their representations to be entirely unavailable.
> >
> > It would help to elaborate on concrete real-world scenarios where even task vectors cannot be preserved, or where retaining them is genuinely infeasible, as this is key to justifying the practical significance of your approach.

---

> > > ### Author Response · Authors · 2025-08-04
> > > **response to reviewer Eirv**
> > >
> > > Thank you for your quick response. Existing model merging approaches face several fundamental limitations and can be revisited from a continual merging perspective. For example,
> > >
> > > 1. In privacy-constrained collaborative scenarios, organizations cannot share complete model parameters but still seek to benefit from collective knowledge.
> > > Consider a scenario where organization A first merges models $\theta^{(1)}$ and $\theta^{(2)}$ to obtain $\theta_{\text{merged}}^{(2)}$, then transmits this merged model to organization B.
> > > Organization B subsequently integrates $\theta_{\text{merged}}^{(2)}$ with their local model $\theta^{(3)}$ to produce $\theta_{\text{merged}}^{(3)}$.
> > > A continual merging approach like OPCM can better mitigate knowledge interference in such sequential scenarios, achieving superior performance compared to conventional batch merging:
> > > $$\mathcal{P}(\text{ContinualMerge}(\theta_{\text{merged}}^{(2)}; \theta^{(0)}, \theta^{(3)})) > \mathcal{P}(\text{BatchMerge}(\theta^{(0)}; \theta_{\text{merged}}^{(2)}, \theta^{(3)}))$$
> > > where $\mathcal{P}(\cdot)$ denotes the performance metric, and ContinualMerge and BatchMerge represent continual and conventional merging algorithms, respectively.
> > > 2. In rapidly evolving domains such as autonomous driving or real-time recommendation systems, continual model merging provides a natural mechanism to *prioritize recent models over older ones*, reflecting the temporal relevance of knowledge.
> > > This temporal weighting is particularly valuable when integrating models that capture evolving data distributions or when merging checkpoints along a training trajectory [1, 2, 3] (when merging along the training trajectory, the number of models to be merged will be large).
> > > For instance, exponential moving average (EMA) based merging can be formulated as:
> > > $$\theta_{\text{EMA}}^{(t)} = \alpha \theta_{\text{EMA}}^{(t-1)} + (1-\alpha) \theta^{(t)}$$
> > > where $\alpha \in [0,1)$ controls the decay rate, ensuring that more recent models $\theta^{(t)}$ receive higher influence while gradually diminishing the contribution of historical knowledge.
> > > This approach is particularly beneficial when merging models along training trajectories, where later checkpoints typically contain more refined representations, or in streaming scenarios where concept drift necessitates adaptive forgetting of outdated patterns.
> > > Unlike conventional merging approaches that treat all models equally, continual merging naturally accommodates such temporal hierarchies and enables dynamic adaptation to evolving requirements.
> > > 3. Some applications require model updates directly on resource-constrained edge devices. These devices have limited storage and memory, making it impractical to store an ever-growing library of past models. A continual merging approach allows the device to integrate a new capability from an incoming model update and then discard the update, maintaining a constant memory footprint while continuously improving.
> > >
> > > [1] Jean Kaddour. Stop Wasting My Time! Saving Days of ImageNet and BERT Training with Latest Weight Averaging.
> > > [2] Sunny Sanyal, et al. Understanding the Effectiveness of Early Weight Averaging for Training Large Language Models.
> > > [3] Li, et al. Model Merging in Pre-training of Large Language Models.

---

> > > > ### Author Response · Authors · 2025-08-04
> > > > **To further illustrate the first point:**
> > > >
> > > > To further illustrate the first point:
> > > > $$\mathcal{P}(\text{ContinualMerge}(\theta_{\text{merged}}^{(2)}; \theta^{(0)}, \theta^{(3)})) > \mathcal{P}(\text{BatchMerge}(\theta^{(0)}; \theta_{\text{merged}}^{(2)}, \theta^{(3)}))$$
> > > > We examine the performance of a state-of-the-art merging method, TSVM [4], under both continual and batch merging settings. The following experimental comparison on an 8-task setting with ViT-B/32 models demonstrates that naively applying a batch-merging algorithm sequentially can lead to performance degradation, highlighting the need for specialized continual merging approaches like ours.
> > > >
> > > > Table: Experimental comparison of TSVM and our method on an 8-task setting.
> > > >
> > > > | Method | Sun397 | Stanford Cars | RESISC45 | EuroSAT | SVHN | GTSRB | MNIST | DTD | Avg. |
> > > > | --- | --- | --- | --- | --- | --- | --- | --- | --- | --- |
> > > > | TSVM (conventional merging) | 67.6 | 71.6 | 84.7 | 93.4 | 91.9 | 92.5 | 98.9 |63.8 | 83.1 |
> > > > |
> > > > | TSVM (continual merging) | 59.9 | 59.5 | 64.0 | 64.7 | 87.9 | 86.1 | 99.5 | 34.3 | 68.5 |
> > > > | **OPCM (Ours, 10 runs)** | 66.2±0.5 | 63.5±0.5 | 78.0±2.2 | 86.5±1.9 | 82.4±1.3 | 74.3±1.4 | 96.9±0.7 | 56.4±1.3 | 75.5±0.5 |
> > > >
> > > > For TSVM under continual merging, we use the task order of SUN397, Stanford Cars, RESISC45, EuroSAT, SVHN, GTSRB, MNIST, and DTD.
> > > > And the update rule is given by:
> > > > $$
> > > > \theta^{(1)}\_{\text{merged}} = \theta^{(1)}, \quad \theta^{(t)}\_{\text{merged}} = \text{TSVM}(\theta^{(t-1)}\_{\text{merged}}, \theta^{(t)}; \theta^{(0)}), \quad t>1.
> > > > $$
> > > >
> > > > [4] Gargiulo et al. , Task Singular Vectors: Reducing Task Interference in Model Merging. CVPR 2025

---

> > > > ### Comment · Reviewer_Eirv · 2025-08-09
> > > >
> > > > My main concern remains whether CMM has practical or potential usefulness. I feel the authors have partially avoided addressing this core issue. While I do not consider CMM to be an important research topic, I will still keep my recommendation as weak accept, as I believe it is worth exploring as an initial attempt in this direction.

---

### Official Review · Reviewer_coXd · 2025-06-16

**Clarity:** 3
**Significance:** 2
**Originality:** 2
**Rating:** 4
**Confidence:** 4

**Summary:**

The authors propose a method for continual model merging, where new models arrive over time and the old models are not available anymore other than via the current merged checkpoint.
The method relies on projections that keep the incoming models' task vector and the current merged model orthogonal as well as adaptive scaling.
The authors show that the method outperforms other baselines for continual model merging on a variety of tasks and provide theoretical analyses of their method.

**Questions:**

- Could you please explain how task information is exactly guaranteed to be preserved in Corollary 5.3? Initially it seems like a trivial rewriting of the merging equation that does not provide any guarantees.

- Could you please expand a bit on the motivation for continual model merging? What are the uses cases where checkpoints could not be kept? I think clarifying this also in the paper could help motivating the method.

- How did you choose the scaling for e.g. Task Arithmetic in Table 6?

**Ethical Concerns:**

["NO or VERY MINOR ethics concerns only"]

**Final Justification:**

The authors have resolved most of my weaknesses, namely, W2-4.
In their response, they provide new experiments and writing as a comparison to prior, very related methods which I believe will find their way into the paper

While I believe that W1 is still true and the task setting is a bit artificial I don't think that it warrants recommending rejecting in itself.
I think it would still be good to address W5, too, which show that there are other methods that can work better in some settings.
In my opinion, this is not necessarily a big weakness in itself but should not be hidden in the Appendix.

Also considering the discussion of the authors with other reviewers I have therefore decided to up my score to 4.

**Limitations:**

The paper discusses some limitations, such as, that the order matters in the method and that SVD adds additional overhead.
However, the limitations could be extended by discussing the results in Appendix E that show that conventional model merging techniques (that are not continual in this works' sense) can outperform the proposed method.

**Quality:**

2

**Strengths And Weaknesses:**

## Strengths

- The paper is easy to read and follow.

- The method proposed by the authors outperforms other baseline techniques for continual model merging on a variety of datasets.

## Weaknesses

- The motivation of memory complexity increasing with the number of models is of course true but this can also be simple disk storage which is easy to get by or even cloud storage which makes the task setting overall a bit artificial.

- I'm not sure if claiming "we present a *novel* perspective on multi-task model merging through the lens of
continual learning" (L96-97) is fair, as the formalization shares many similarities with Dziadzio et al. 2024, which is nevertheless also discussed and cited.

- The adaptation of Task Arithmetic in Eq. 5 is not consistent with theoretical derivations of Task Arithmetic, e.g., from Daheim et al. 2024, as the task vector is calculated wrt. a different model than the one it is added too, which could be discussed more.

- The work should discuss other papers that use SVD to reduce Task interference more, such as Gargiulo et al. 2025.  Perhaps a comparison could also be added.

- The fact that the method is outperformed by other merging techniques like RegMean if the restriction on continually merging and not keeping all checkpoints is lifted is hidden in Appendix E which is not referenced in the main paper. It would be more appropriate to discuss and reference it there.

## References

Daheim, Nico, et al. "Model merging by uncertainty-based gradient matching." ICLR, 2024.

Gargiulo, Antonio Andrea, et al. "Task singular vectors: Reducing task interference in model merging." Proceedings of the Computer Vision and Pattern Recognition Conference. 2025.

---

> ### Author Rebuttal · Authors · 2025-07-29
>
> We thank the reviewer for their thoughtful feedback and constructive suggestions.
>
> ## Response to weakness 1 and question 2
>
> > **W1:** The motivation of memory complexity increasing with the number of models is of course true but this can also be simple disk storage which is easy to get by or even cloud storage which makes the task setting overall a bit artificial.
>
> > **Q2:** Could you please expand a bit on the motivation for continual model merging? What are the uses cases where checkpoints could not be kept? I think clarifying this also in the paper could help motivating the method.
>
> While disk storage is indeed available, the continual model merging setting is motivated by practical situations where models are frequently updated—often by different teams or individuals—and earlier versions may not be retained or accessible.
>
> Besides, the $O(T|\theta|)$ complexity isn't just about storage, it also affects the computational overhead during merging.
>
> ## Response to weakness 2
>
> > I'm not sure if claiming "we present a novel perspective on multi-task model merging through the lens of continual learning" (L96-97) is fair, as the formalization shares many similarities with Dziadzio et al. 2024, which is nevertheless also discussed and cited.
>
> We appreciate the reviewer's comment, and we acknowledge that our work shares some similarities with Dziadzio et al. 2024 ("How to Merge Your Multimodal Models Over Time?").
> However, our work presents a distinct perspective on continual model merging and framework, particularly in the following ways:
>
> 1. The framework of Dziadzio et al. 2024 is different from ours, as shown in the following:
> ```
> Dziadzio et al. 2024 (models are continually trained, and the set of models is growing over time):
>
> a set of models  ---(append)--> new set of models --(append)--> new set of models ...
>     🡫                 🡡              🡫                🡡.           🡫
>   merge-1 -(train)-> task-1        merge-2 -(train)-> task-2       merge-3 -(train)-> task-3 ...
> ```
> ```
> Ours (all models are independently trained and merged in a continual manner):
> pre-trained --🡫--------🡫---------🡫 ...
>             task-1  -> task-2  -> task-3 ...
>               🡫        🡫         🡫
>             merge-1 -> merge-2 -> merge-3 ...
> ```
> - In Dziadzio et al. 2024, the set of models is continually growing over time, and the merging is performed on the entire set of models at each step.
> - In our work, the models are independently trained and merged in a continual manner, where each task is merged with the previous tasks' merged model.
> 2. The theoretical analysis in Dziadzio et al. 2024 and ours are focused on different aspects:
>    - Dziadzio et al. 2024 focus on the initialization of the merging operation, i.e., how to select the set of models to be merged.
>    - In our work, we focus on analyzing the merging operation itself, i.e,. how to reduce task interference during merging. We propose a projection-based continual model merging method that ensures the task vectors remain orthogonal, which is a key aspect of our contribution.
> 3. What's more, Dziadzio et al. 2024  is a contemporaneous work; at the time of writing our manuscript, it was not yet available. We have included a discussion of this work in a revised version of the manuscript.
>
> ## Response to weakness 3
>
> > The adaptation of Task Arithmetic in Eq. 5 is not consistent with theoretical derivations of Task Arithmetic, e.g., from Daheim et al. 2024, as the task vector is calculated wrt. a different model than the one it is added too, which could be discussed more.
>
> Eq.(5) in our manuscript  is equivalent to the original Task Arithmetic formulation, as shown below:
> $$
> \theta\_{\text{merged}}^{(t)} = \theta\_{\text{merged}}^{(t-1)} + \lambda (\theta^{(t)} - \theta^{(0)}), \quad Eq.(5) (\text{continual task arithmetic})
> $$
> and the original Task Arithmetic formulation is:
> $$
> \theta_{\text{merged}} = \theta_{\text{0}} + \lambda \sum_{i=1}^{t} (\theta^{(i)} - \theta^{(0)}),
> $$
> where $\theta^{(0)}$ is the pre-trained weights.
>
> *proof:*
> We prove the equivalence by induction:
> 1. For the first task, we have:
> $$\theta_{\text{merged}}^{(1)} = \theta_{\text{0}} + \lambda (\theta^{(1)} - \theta^{(0)}).$$
> 2. Assume the formula holds for $t-1$ tasks, i.e.,
> $$\theta_{\text{merged}}^{(t-1)} = \theta_{\text{0}} + \lambda \sum_{i=1}^{t-1} (\theta^{(i)} - \theta^{(0)}).$$
> 3. According to Eq.(5), we have:
> $$\theta_{\text{merged}}^{(t)} = \theta_{\text{merged}}^{(t-1)} + \lambda (\theta^{(t)} - \theta^{(0)}) = \theta_{\text{0}} + \lambda \sum_{i=1}^{t-1} (\theta^{(i)} - \theta^{(0)}) + \lambda (\theta^{(t)} - \theta^{(0)}) = \theta_{\text{0}} + \lambda \sum_{i=1}^{t} (\theta^{(i)} - \theta^{(0)}).$$
> Thus, by induction, the formula holds for all $t$ tasks.
>
> ## Response to weakness 4
>
> > The work should discuss other papers that use SVD to reduce Task interference more, such as Gargiulo et al. 2025. Perhaps a comparison could also be added.
>
> Thank you for this valuable feedback. Gargiulo et al. 2025 ("Task Singular Vectors: Reducing Task Interference in Model Merging") introduce task singular vectors merging (TSVM) that addresses task interference in model merging through SVD decomposition. TSVM reduces task interference by whitening the TSVs after SVD decomposition across all tasks simultaneously. In contrast, our method reduces interference by projecting new task vectors orthogonally to prior merged vectors at each step. Below is an experimental comparison conducted on the 8-task setting using ViT-B/32 models:
>
> Table: Experimental comparison of TSVM and our method on an 8-task setting.
>
> | Method | Sun397 | Stanford Cars | RESISC45 | EuroSAT | SVHN | GTSRB | MNIST | DTD | Avg. |
> | --- | --- | --- | --- | --- | --- | --- | --- | --- | --- |
> | TSVM (conventional merging) | 67.6 | 71.6 | 84.7 | 93.4 | 91.9 | 92.5 | 98.9 |63.8 | 83.1 |
> |
> | TSVM (continual merging) | 59.9 | 59.5 | 64.0 | 64.7 | 87.9 | 86.1 | 99.5 | 34.3 | 68.5 |
> | **OPCM (Ours, 10 runs)** | 66.2±0.5 | 63.5±0.5 | 78.0±2.2 | 86.5±1.9 | 82.4±1.3 | 74.3±1.4 | 96.9±0.7 | 56.4±1.3 | 75.5±0.5 |
>
> For TSVM under continual merging, we use the task order of SUN397, Stanford Cars, RESISC45, EuroSAT, SVHN, GTSRB, MNIST, and DTD.
> And the update rule is given by:
> $$
> \theta^{(1)}\_{\text{merged}} = \theta^{(1)}, \quad \theta^{(t)}\_{\text{merged}} = \text{TSVM}(\theta^{(t-1)}\_{\text{merged}}, \theta^{(t)}; \theta^{(0)}), \quad t>1.
> $$
>
> ## Response to question 1
>
> > Could you please explain how task information is exactly guaranteed to be preserved in Corollary 5.3? Initially it seems like a trivial rewriting of the merging equation that does not provide any guarantees.
>
> $$
> W^{(t)}\_{\text{merged}} - W^{(0)} = \frac{1}{\lambda(t)} \sum_{i=1}^{t} P^{(i-1)}\_{\alpha}(\Delta W^{(i)}), \quad(\text{Corollary 5.3})
> $$
> This corollary guarantees that when a new task-specific model is merged, *it doesn't overwrite the subspace spanned by the old tasks*, their contributions are geometrically independent.
>
> ## Response to question 3
>
> > How did you choose the scaling for e.g. Task Arithmetic in Table 6?
>
> We performed a grid search over the scaling factor for Task Arithmetic, evaluating values in [0.1, 0.2, 0.3, 0.4, 0.5, 0.6, 0.7, 0.8, 0.9] on the test set of each task.
> The scaling factor is selected based on the best average performance across all tasks, which is 0.3 for the eight-experiment setting in Table 6.

---

> ### Author Response · Authors · 2025-08-06
> **Hi**
>
> Dear Reviewer,
>
> As the rebuttal period is coming to an end, we would like to kindly ask whether our responses have addressed your concerns. Please let us know if there are any remaining questions or clarifications we can provide. We’re happy to respond further if needed.

---

> > ### Comment · Reviewer_coXd · 2025-08-07
> > **Thank you for your response!**
> >
> > Dear authors,
> >
> > thank you for your response!
> > I don't have any open questions left for now.
> >
> > While I would like to point out that the call for papers states "March 1st, 2025" as a cut-off for contemporaneous work (https://neurips.cc/Conferences/2025/CallForPapers) I think your response has addressed W2 and W3,4 and the results in your response to W4 seem convincing.
> >
> > While I still think the task set-up is a bit artificial and keeping models is usually not a problem (W1), also for the merging operation, this should not take away from the contribution itself.
> >
> > I will revise my score based on your response.

---

> > > ### Author Response · Authors · 2025-08-07
> > > **Thank you for your feedback!**
> > >
> > > Thanks again for your insightful feedback! We will carefully organize the content of the rebuttal in accordance with your suggestions.

---

### Official Review · Reviewer_kfjD · 2025-06-24

**Clarity:** 3
**Significance:** 4
**Originality:** 4
**Rating:** 4
**Confidence:** 4

**Summary:**

The paper introduces OPCM, a training‑free, continual model merging method. It sequentially integrates new fine‑tuned models into a merged model via orthogonal projections and adaptive scaling, reducing interference and maintaining constant memory per merge.

**Questions:**

- 1. In classification settings, the method does not consider the classification head, and the tasks used appear to be relatively similar. As a result, for Continual Model Merging, there is a possibility that only a few models dominate the merged result, leading to low variance and insensitivity to task order.
   - Does the use of simple classification tasks limit the persuasiveness of the method for Continual Model Merging?
   - Would it be more meaningful to evaluate the method on more diverse tasks with shared objectives, such as language generation, where the differences between tasks are more pronounced?
- 2. Why is the accuracy of the single EuroSAT task in Figure 12(a) significantly lower than the accuracy when MNIST and EuroSAT are merged?

**Ethical Concerns:**

["NO or VERY MINOR ethics concerns only"]

**Final Justification:**

The authors addressed my concerns, but limitations raised by other reviewers remain. I keep my score: boardline accept.

**Limitations:**

yes

**Quality:**

4

**Strengths And Weaknesses:**

Strengths:
- 1. The paper is clearly written and addresses a novel and timely problem in model merging.

- 2. The authors conduct comprehensive experiments to demonstrate the effectiveness of their method.

Weaknesses:
- 1. In Table 1, the results for C. finetune lack variance/error bars.
- 2. The extremely low variance of accuracy when using SWA in Table 1 suggests the tasks are highly similar, and possibly only a few models dominate the merged performance. To strengthen the claim of order-robustness, the authors should consider testing on tasks with greater diversity.

---

> ### Author Rebuttal · Authors · 2025-07-27
>
> We thank the reviewer for the thoughtful feedback and suggestions.
>
> ## Response to weakness 1
>
> > In Table 1, the results for C. finetune lack variance/error bars.
>
> We appreciate the reviewer's attention to statistical rigor. We chose not to include variance/error bars for continual fine-tuning (C. finetune) due to computational constraints rather than oversight.
>
> Continual fine-tuning is extremely computationally intensive. For instance, in our 20-task experiments, each task requires 20-30 minutes of fine-tuning. A single complete run, therefore, takes 8-10 hours. To obtain meaningful variance estimates, we would need multiple runs across different task orderings and random seeds.
>
> Given that we evaluate three different model sizes (ViT-B/32, ViT-B/16, ViT-L/14) across multiple task configurations (8, 14, and 20 tasks), computing variance for continual fine-tuning would require several months (about 3-5 months for 10 runs) of continuous GPU time. This is prohibitively expensive for our current resources.
>
> Although we report continual fine-tuning results from single runs, the comparison with our method's statistical analysis reveals compelling insights:
>
> 1. In 5 out of 9 experimental settings (highlighted in **bold**), even our method's minimum performance exceeds continual fine-tuning's single-run results.
> 2. In the remaining cases, our method still achieves competitive results within 1-5% of continual fine-tuning.
> 3. OPCM achieves the performance with orders of magnitude lower computational cost, requiring only a few seconds versus 20-30 minutes per task for continual fine-tuning.
>
> Table: Comparison of Continual Fine-tuning and OPCM Results.
>
> | Experimental Setup | Continual fine-tuning Avg Acc. | OPCM Avg Acc.(10-runs) |
> | --- | --- | --- |
> | ViT-B/32, 8 tasks  | 79.8 | 75.5$\pm$0.5, Min: 74.8, Max:76.5  |
> | ViT-B/16, 8 tasks | 82.9 | 81.8$\pm$0.3, Min: 81.3, Max: 82.3 |
> | ViT-L/14, 8 tasks | 90.0 | 87.0$\pm$0.4, Min: 86.6, Max: 87.7 |
> | ViT-B/32, 14 tasks | 67.4 | 71.9$\pm$0.3, Min: **71.3**, Max: 72.5 |
> | ViT-B/16, 14 tasks | 72.2 | 77.1$\pm$0.5, Min: **76.2**, Max: 77.8 |
> | ViT-L/14, 14 tasks | 70.9 | 83.5$\pm$0.2, Min: **83.1**, Max: 83.8 |
> | ViT-B/32, 20 tasks | 62.6 | 65.7$\pm$0.2, Min: **65.3**, Max: 66.0 |
> | ViT-B/16, 20 tasks | 68.2 | 70.3$\pm$0.2, Min: **69.8**, Max: 70.7 |
> | ViT-L/14, 20 tasks | 77.7 | 76.0$\pm$0.2, Min: 75.6, Max: 76.4 |
>
> ## Response to weakness 2
>
> > The extremely low variance of accuracy when using SWA in Table 1 suggests the tasks are highly similar, and possibly only a few models dominate the merged performance. To strengthen the claim of order-robustness, the authors should consider testing on tasks with greater diversity.
>
> The extremely low variance observed with SWA in Table 1 is due to the mathematical equivalence of SWA and simple averaging, which we clarify below.
> The equivalence between SWA and simple averaging implies that SWA is inherently order-agnostic, as the simple average does not depend on the order of the models being processed.
>
> **Proof of equivalence of SWA and simple averaging:**
> Let's denote the parameters of task-specific models as $\theta^{(1)}, \theta^{(2)}, \ldots, \theta^{(n)}$ for $n$ tasks.
> For stochastic weight averaging (SWA), the update rule is given by:
> $$
>   \theta_{\text{SWA}}^{(1)} = \theta^{(1)}, \quad \theta_{\text{SWA}}^{(t)} = \frac{\theta_{\text{SWA}}^{(t-1)}(t-1) + \theta^{(t)}}{t},
> $$
> We processed by induction:
> For the first task, the SWA update rule gives;
> $$
>     \theta_{\text{SWA}}^{(1)} = \theta^{(1)}.
> $$
> Assume the formula holds for $t-1$ tasks, i.e.,
> $$
>     \theta_{\text{SWA}}^{(t-1)} = \frac{1}{t-1} \sum_{i=1}^{t-1} \theta^{(i)}.
> $$
> According to the SWA update rule, we have:
> $$
>     \theta_{\text{SWA}}^{(t)} = \frac{\theta_{\text{SWA}}^{(t-1)}(t-1) + \theta^{(t)}}{t} = \frac{(t-1) \cdot \frac{1}{t-1} \sum_{i=1}^{t-1} \theta^{(i)} + \theta^{(t)}}{t} = \frac{1}{t}
>     \sum_{i=1}^{t} \theta^{(i)}.
> $$
> Thus, by induction, the SWA update rule holds for all $t$ tasks.
>
> **Similarity of selected tasks:**
> The tasks used in our experiments are: SUN397, Stanford-Cars, RESISC45, EuroSAT, SVHN, GTSRB, MNIST, DTD, Oxford_Flowers102, PCAM, FER2013, Oxford-IIIT-Pet, STL10, CIFAR100, CIFAR10, Food101, Fashion_MNIST, EMNIST_Letters, KMNIST, and Rendered-SST2.
> These tasks span a wide range of domains, including:
>
> - Natural scene classification: SUN397
> - Fine-grained object recognition: Stanford-Cars, Oxford_Flowers102
> - Remote sensing: RESISC45, EuroSAT
> - Digit and character recognition: MNIST, SVHN, EMNIST_Letters, and KMNIST
> - Traffic sign recognition: GTSRB
> - Texture classification: DTD
> - Medical imaging: PCAM
> - Facial expression recognition: FER2013
> - Pet breed classification: Oxford-IIIT-Pet
> - General object recognition: CIFAR10, CIFAR100, STL10, Food101
> - Fashion product classification: Fashion_MNIST
> - Sentiment analysis: Rendered-SST2
>
> As shown in Figures 4,8,9, and Figure 10, the cosine similarity between task vectors is generally low, empirically indicating that the tasks are indeed diverse and the learned representations are approximately orthogonal.
>
> ## Response to question 1
>
> > In classification settings, the method does not consider the classification head, and the tasks used appear to be relatively similar. As a result, for Continual Model Merging, there is a possibility that only a few models dominate the merged result, leading to low variance and insensitivity to task order.
> >
> > - Does the use of simple classification tasks limit the persuasiveness of the method for Continual Model Merging?
> > - Would it be more meaningful to evaluate the method on more diverse tasks with shared objectives, such as language generation, where the differences between tasks are more pronounced?
>
> **Classification head in experiments:**
> In our experiments, we follow the common practice in model merging literature of using zero-shot classification heads initialized with the pre-trained CLIP model's text encoder. Specifically, we construct task-specific classification heads using CLIP's text encoder to generate zero-shot weights for each task's class names and templates.
> When fine-tuning the vision encoder, the classification head remains fixed. This is meant to preserve the open-vocabulary nature of the CLIP model, so it can still handle unseen classes during inference.
>
> **On task similarity and robustness:**
> The tasks we selected for our experiments are diverse, covering various domains such as natural scene classification, as clarified above in the response to weakness 2.
>
> On the other hand, as shown in Figures 4,8,9, and Figure 10, the cosine similarity between task vectors is generally low, empirically indicating that the tasks are indeed diverse and the learned representations are approximately orthogonal.
>
> From the experiments, we observe that (1) the task vectors are roughly orthogonal (Figure 4 and Figure 10), and (2) the proposed method remains robust to task order (Table 1 and Figure 5).
>
> **Experimental results on language generation tasks:**
>
> We conduct experiments on language generation tasks on Flan-T5-base models on eight tasks from the GLUE benchmark. The results are shown below:
>
> Table: Results on Language Generation Tasks
>
> | Method | CoLA | MNLI | MRPC | QNLI | QQP | RTE | SST2 | STSB | Avg.$\uparrow$ | BWT$\uparrow$ |
> | --- | --- | --- | --- | --- | --- | --- | --- | --- | --- | --- |
> | Fine-tuned | 69.1 | 82.7 | 85.5 | 90.9 | 84.0 | 84.4 | 92.9 | 87.4 | 84.6 | - |
> |
> | Averaging (SWA) | 69.1 | 62.6 | 79.4 | 89.8 | **83.9** | 81.2 | 91.7 | 73.2 | 78.8$\pm$0.0 | -2.5$\pm$1.1 |
> | Continual task arithmetic | 70.5 | 57.8 | 78.4 | **90.2** | 83.6 | 80.5 | **92.3** | 77.8 | 78.9$\pm$0.0 | -1.8$\pm$1.1 |
> | Continual ties-merging | 68.8 | 50.5 | **79.8** | 89.9 | 83.1 | 79.2 | 91.9 | **79.3** | 77.8$\pm$1.6 | -0.5$\pm$1.9 |
> | **OPCM(Ours)** | **69.7** | **72.9** | 78.8 | **90.2** | 83.8 | **82.2** | **92.3** | 74.7 | **80.6**$\pm$0.3 | -2.1$\pm$0.6 |
>
> The results show that OPCM outperforms the other continual merging methods on average performance across the eight tasks.
>
> ## Response to question 2
>
> > Why is the accuracy of the single EuroSAT task in Figure 12(a) significantly lower than the accuracy when MNIST and EuroSAT are merged?
>
> I suppose the reviewer is referring to Figure 11(a) instead of Figure 12(a).
> If there is a misunderstanding, please let us know.
>
> In Figure 11, each row represents a "new task" being merged in the continual merging process, and each column represents a "test task" being evaluated.
> The diagonal entries show the performance of each task when evaluated on itself, and the off-diagonal entries show cross-task performance (previous tasks).
>
> For example, in Figure 11(a), the first task is SUN397 (at t=1), we evaluate the performance of the merged model ($\theta\_{merged}^{(1)}$) on SUN397.
> For $t>1$, the merged model ($\theta\_{merged}^{(t)} =ContinualMerging(\theta\_{merged}^{(t-1)}; \theta^{(0)}, \theta^{(t)})$, where $\theta^{(0)}$ is the pre-trained model), is evaluated on the current task and all previous tasks.
>
> Table: Test Accuracy Matrix for ViT-B/32 on 8 Tasks (Figure 11a)
>
> | **step** | New Task | SUN397 | GTSRB | Cars | DTD | RESISC45 | SVHN | MNIST | EuroSAT |
> |---|---|---|---|---|---|---|---|---|---|
> |t=1| SUN397 | 0.75 | -     | -    | -   | -        | -    | -     | -       |
> |t=2| GTSRB | 0.70 | 0.96  | -    | -   | -        | -    | -     | -       |
> |t=3| Cars | 0.69 | 0.87  | 0.69 | -   | -        | -    | -     | -       |
> |t=4| DTD | 0.68 | 0.79  | 0.68 | 0.58| -        | -    | -     | -       |
> |t=5| RESISC45 | 0.67   | 0.67  | 0.66 | 0.55| 0.77     | -    | -     | -       |
> |t=6| SVHN | 0.67 | 0.64  | 0.65 | 0.53| 0.74     | 0.70 | -     | -       |
> |t=7| MNIST | 0.66 | 0.59  | 0.64 | 0.52| 0.71     | 0.70 | 0.89  | -       |
> |t=8| EuroSAT | 0.65 | 0.55  | 0.62 | 0.51| 0.71     | 0.65 | 0.86  | 0.76    |

---

> > ### Comment · Reviewer_kfjD · 2025-08-04
> > **Official Comment by Reviewer kfjD**
> >
> > Thanks for the response. I keep my score as is.

---

### Official Review · Reviewer_2FXy · 2025-07-01

**Clarity:** 3
**Significance:** 2
**Originality:** 3
**Rating:** 4
**Confidence:** 3

**Summary:**

The authors tackle the problem of merging fine-tuned models in a sequential setting where models arrive over time without retraining or revisiting data. Unlike prior approaches that rely on simultaneous access to all models and simple weight averaging, the authors propose a projection-based method that merges each new model by projecting its updates orthogonally to the subspace of previously merged updates, with an adaptive scaling mechanism to control drift. This setup helps mitigate task interference and maintain performance across tasks. The method is simple, memory-efficient (constant in the number of models), and shows strong empirical results on CLIP-ViT, outperforming continual baselines by 5–8% on average

**Questions:**

How is this different than the works I listed about from the orthogonality contribution.

In addition, what about this paper:

https://arxiv.org/html/2505.12082v1

**Ethical Concerns:**

["NO or VERY MINOR ethics concerns only"]

**Final Justification:**

I think due to the lack of clear novelty or strong empirical results. I decide to hold my score.

**Limitations:**

Does having the weights be projected orthogonal to each other cause limitations - lack of sharing useful information from different tasks?

**Quality:**

3

**Strengths And Weaknesses:**

What stands out in this paper is how it reframes model merging as a continual process where models arrive one at a time rather than being merged all at once. That feels like a much more realistic and practical setting, especially for real-world systems where tasks evolve over time. The idea is clean and well-motivated. The method itself is simple, does not require retraining, and keeps memory use low, which makes it easy to apply. The results are strong and consistent across model sizes and task counts, and it's good to see that performance holds up even when task order changes.


The technical novelty is on the lighter side. Using orthogonal projections to reduce interference has been done before in the continual learning literature. Adaptive scaling makes sense in this context but is also a pretty natural move. The method is well-executed, but it mostly repurposes existing ideas for the merging setting rather than introducing something fundamentally new. It would help to better understand what is specific to merging here that is not already covered in related projection-based continual learning approaches.

https://arxiv.org/abs/2301.12131
https://openreview.net/pdf?id=hac6DzbMa7

---

> ### Author Rebuttal · Authors · 2025-07-25
>
> We thank the reviewer for the thoughtful feedback.
> We will revise the manuscript to include the following clarifications on the points raised, and cite the relevant works listed by the reviewer.
>
> ## Response to weaknesses: distinction from existing projection-based continual learning methods
>
> > Using orthogonal projections to reduce interference has been done before in the continual learning literature. Adaptive scaling makes sense in this context but is also a pretty natural move. The method is well-executed, but it mostly repurposes existing ideas for the merging setting rather than introducing something fundamentally new. It would help to better understand what is specific to merging here that is not already covered in related projection-based continual learning approaches. [1,2]
>
> While orthogonal projections have been used in continual learning, our approach differs fundamentally from existing methods in the following ways:
>
> Projection of gradients in continual learning [1,2] is different from the projection operation in the proposed method in their objectives and mechanisms.
>
> 1. In OPCM (our proposed method), we focus on combining multiple fine-tuned models, each specialized for a specific task, into a single multi-task model. The projection in OPCM operates on the parameter deltas (task vectors) between a fine-tuned model and the base pre-trained model. In contrast, projection-based continual learning methods are used during the training process of a single model to learns multiple tasks sequentially.
> 2. The projection in OPCM is performed after the fine-tuning of each model, while continual learning methods apply projections during the training process, which requires access to the original training data and is computationally expensive.
> 3. For projection-based continual learning methods such as OGD, due to the requirement of storing the gradients of previous tasks, the memory complexity is $O(Td)$, where $T$ is the number of tasks and $d$ is the size of the model. In contrast, OPCM maintains a single merged model, which represents the cumulative results of merging the pre-trained model with task-specific models up to task $t$. The memory complexity of OPCM is $O(d)$.
> 4. The computational mechanism of projection in OPCM is different from that in projection-based continual learning methods. In OPCM, we use SVD on the accumulated merged weights $\Delta W_{\text{merged}}^{(t-1)}$ to identify the important subspace, and then project the new task vector $\Delta W_t^{(t)}$ orthogonally to this subspace to avoid interference with previously learned tasks.
> The projected weights are computed as:
> $$\mathcal{P}^{(t-1)}\_\alpha\left(\Delta W^{(t)}\right) = \sum\_{i,j=r_{\alpha}, i\neq j}^{m, n} {\left\langle \Delta W^{(t)}, u_i v_j^T \right\rangle}\_F u_i v_j^T,$$
> In contrast, projection-based continual learning methods typically rely on the stored gradients or other representations from previous tasks.
> The projected gradients are computed as:
> $$g_t - \mathcal{P}\_{\text{span}(g_1, \ldots, g_{t-1})}(g_t)$$
>
> | Aspect | Continual Learning Methods | Our Method (OPCM) |
> | --- | --- | --- |
> | Setting | Model Merging | Continual Learning |
> | Operation | Gradients during training | Post-training, task vectors during merging |
> | Requirements | Training data required | No data access and training-free |
> | Memory | $O(Td)$, where $T$ is the number of tasks | $O(d)$, constant |
> | Mechanism | Uses SVD on the accumulated merged task vector to identify subspace | Orthogonal to past gradients or subspaces |
>
> ## Response to questions: how is this different from [3]
>
> > How is this different than the works I listed about from the orthogonality contribution.
> >
> > In addition, what about this paper:
> >
> > https://arxiv.org/html/2505.12082v1 [3]
>
> The proposed OPCM merges fine-tuned models on different downstream tasks after the training phase, which is entirely post-training. While the method introduced in [3] applies model merging during pre-training, requiring access to sequential checkpoints from the training trajectory.
>
> Another key difference between these two settings is that OPCM merges fine-tuned models post-training using orthogonal projections to preserve task-specific performance, limiting direct task-specific information sharing to avoid interference.
> While for model merging in the pre-training phase, such as simple moving averaging ($M_{avg} = \frac{1}{N}\sum_{i=1}^{N} M_i$) and exponential moving averaging ($M_{avg}^{(i)} = \alpha M_i + (1 - \alpha) M_{avg}^{(i-1)}$), sequential checkpoints are from the same training trajectory, the divergence between the models is not as large as that for task-specific model merging, thus allowing smoother integration of knowledge across checkpoints.
> Averaging during pre-training smooths the parameter trajectory, which can enhance the robustness and generalization by reducing overfitting to specific training iterations or data batches.
>
> ## Response to limitation: The limitation of orthogonal projection
>
> > Does having the weights be projected orthogonal to each other cause limitations - lack of sharing useful information from different tasks?
>
> Yes, projecting weights to be orthogonal to each other, as done in OPCM, can sometimes cause limitations in the direct sharing of useful information across tasks. However, this limitation is a deliberate trade-off that brings advantages, such as preventing interference and reducing knowledge redundancy.
>
> 1. **Reduced direct information sharing**: In OPCM, the goal is to merge multiple pre-trained models sequentially while preserving performance on earlier tasks. To achieve this, task-specific updates (or task vectors) are projected onto a subspace orthogonal to the important parameter directions of previously merged models.
> This orthogonality means that a new task cannot adjust the model’s parameters in ways that overlap with or build upon the protected directions of prior tasks. For example, if Task A learns a feature that could also benefit Task B, Task B’s updates cannot directly modify or enhance that feature if it lies in a protected direction. As a result, the model may miss opportunities to leverage task-specific knowledge from one task to improve performance on another.
>
> 2. **Preventing knowledge redundancy**: While orthogonality imposes this limitation, it also offers a significant advantage: it prevents knowledge redundancy. In a non-orthogonal approach, different tasks might encode similar features or information in their updates, duplicating effort and wasting the model’s capacity.
> For instance, if two tasks require recognizing similar low-level patterns, they might redundantly adjust parameters to represent that knowledge independently.
> In OPCM, the base model provides a foundation of shared features accessible to all tasks, and the orthogonal projection ensures that each task’s updates are unique and keep the merged model lying in a low-loss basin that is common to all tasks.
> By avoiding duplication, OPCM helps maintain the model’s generalization ability across tasks and prevents it from being pushed into a suboptimal region of the parameter space.
>
> To summarize, by enforcing orthogonality, OPCM minimizes interference, knowledge redundancy, and catastrophic forgetting, helping to ensure that adding a new task does not significantly degrade performance on earlier ones.
>
> [1] Yang et al. - 2023 - Restricted Orthogonal Gradient Projection for Continual Learning.
> [2] Wang et al. - 2023 - Continual Learning with Orthogonal Weights and Knowledge Transfer.
> [3] Li et al. - 2025 - Model Merging in Pre-training of Large Language Models.

---

> > ### Comment · Reviewer_2FXy · 2025-08-05
> >
> > Thank you for the clarifications, I decide to keep my score.

---

### Decision · Program_Chairs · 2025-09-17

**Decision:**

Accept (poster)

**Comment:**

The paper introduces a continual model merging setting where fine tuned models arrive sequentially and speed of merging and storage are factors. They propose a method based on orthogonal projects with robustness of results to the task order. Experiments highlight the benefits in this setting. While reviewers noted some limited novelty in the projection based method they agreed the formulation of continual model merging is relevant and the experiments well validated